# Unsupervised Learning of Group Invariant and Equivariant Representations

**Robin Winter**\*
Bayer AG
Freie Universität Berlin
robin.winter@bayer.com

**Marco Bertolini**\*
Bayer AG
marco.bertolini@bayer.com

**Tuan Le**
Bayer AG
Freie Universität Berlin
tuan.le2@bayer.com

**Frank Noé**
Freie Universität Berlin
Microsoft Research
frank.noe@fu-berlin.de

**Djork-Arné Clevert**
Bayer AG
djork-arne.clevert@bayer.com

## Abstract

Equivariant neural networks, whose hidden features transform according to representations of a group $G$ acting on the data, exhibit training efficiency and an improved generalisation performance. In this work, we extend group invariant and equivariant representation learning to the field of unsupervised deep learning. We propose a general learning strategy based on an encoder-decoder framework in which the latent representation is separated in an invariant term and an equivariant group action component. The key idea is that the network learns to encode and decode data to and from a group-invariant representation by additionally learning to predict the appropriate group action to align input and output pose to solve the reconstruction task. We derive the necessary conditions on the equivariant encoder, and we present a construction valid for any $G$, both discrete and continuous. We describe explicitly our construction for rotations, translations and permutations. We test the validity and the robustness of our approach in a variety of experiments with diverse data types employing different network architectures.

## 1 Introduction

An increasing body of work has shown that incorporating knowledge about underlying symmetries in neural networks as inductive bias can drastically improve the performance and reduce the amount of data needed for training Cohen & Welling (2016a); Bronstein et al. (2021). For example, the equivariant design with respect to the translation symmetry of objects in images proper of convolutional neural networks (CNNs) has revolutionized the field of image analysis LeCun et al. (1995). Message Passing neural networks, respecting permutation symmetries in graphs, have enabled powerful predictive models on graph-structured data Gilmer et al. (2017); Defferrard et al. (2016). Recently, much work has been done utilizing 3D rotation and translation equivariant neural networks for point clouds and volumetric data, showing great success in predicting molecular ground state energy levels with high fidelity Miller et al. (2020); Anderson et al. (2019); Klicpera et al. (2020); Schütt et al. (2021). Invariant models take advantage of the fact that often properties of interest, such as the class label of an object in an image or the ground state energy of a molecule, are invariant to certain group actions (e.g., translations or rotations), while the data itself is not (e.g., pixel values, atom coordinates).

There are several approaches to incorporate invariance into the learned representation of a neural network. The most common approach consists of teaching invariance to the model by data augmenta-

---

\*equal contribution

36th Conference on Neural Information Processing Systems (NeurIPS 2022).

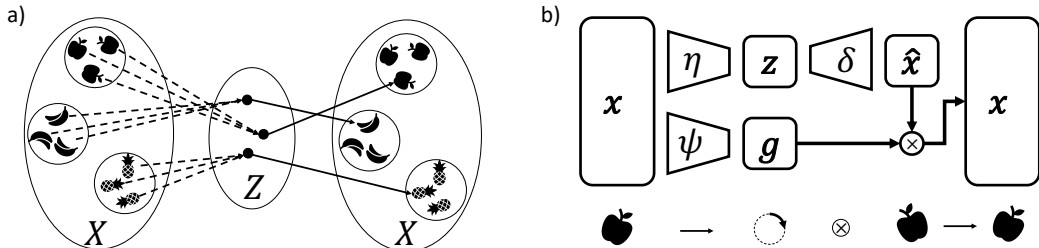

Figure 1: a) Schematic of the learning task this work is concerned with. Data points $x \in X$ are encoded to and decoded from latent space $Z$. Points in the same orbit in $X$ are mapped to the same point (orbit) $z \in Z = X/G$. Latent points $z$ are mapped to canonical elements $\hat{x} \in \{\rho_X(g)x | \forall g \in G\}$. b) Schematic of our proposed framework with data points $x$, encoding function $\eta$, decoding function $\delta$, canonical elements $\hat{x}$, group function $\psi$ and group action $g$

tion: during training, the model must learn that a group transformation on its input does not affect its label. While this approach can lead to improved generalization performance, it reduces training efficiency and quickly becomes impractical for higher dimensional data Thomas et al. (2018). A second technique, known as feature averaging, consists of averaging model predictions over group transformations of the input Puny et al. (2021). While feasible with finite groups, this method requires, for instance, sampling for infinite groups Lyle et al. (2020). A third approach is to impose invariance as a model architectural design. The simplest option is to restrict the function to be learned to be a composition of symmetric functions only Schütt et al. (2018). Such choice, however, can significantly restrict the functional form of the network. A more expressive variation of this approach consists of an equivariant neural network, followed by a symmetric function. This allows the network to leverage the benefits of invariance while having a larger capacity due to the less restrictive nature of equivariance. In fact, in many real-world application, equivariance is beneficial if not necessary Smidt (2020); Miller et al. (2020). For example, the interaction of a molecule (per se rotational invariant) with an external magnetic field is an intrinsically equivariant problem.

All aforementioned considerations require some sort of supervision to extract invariant representations from data. Unsupervised learning of group invariant representations, despite its potential in the field of representation learning, has been impaired by the fact that the representation of the data in general does not manifestly exhibit the group as a symmetry. For instance, in the case of an encoder-decoder framework in which the bottleneck layer is invariant, the reconstruction is only possible up to a group transformation. Nevertheless, the input data is typically parametrized in terms of coordinates in some vector space $X$, and the reconstruction task can only succeed by employing knowledge about the group action on $X$.

Following this line of thought, this work is concerned with the question: *Can we learn to extract both the invariant and the (complementary) equivariant representations of data in an unsupervised way?*

To this end, we introduce a group-invariant representation learning method that encodes data in a group-invariant latent code and a group action. By separating the embedding in a group-invariant and a group-equivariant part, we can learn expressive lower-dimensional group-invariant representations utilizing the power of autoencoders (AEs). We can summarize the main contributions of this work as follows:

- We introduce a novel framework for learning group equivariant representations. Our representations are *by construction* separated in an invariant and equivariant component.

- We characterize the mathematical conditions of the group action function component and we propose an explicit construction suitable for *any* group $G$. To the best of our knowledge, this is the first method for unsupervised learning of separated invariant-equivariant representations valid for any group.

- We show in various experiments the validity and flexibility of our framework by learning representations of diverse data types with different network architectures. We also show that the invariant representations are superiour to the non-invariant counterparts in downstream tasks, and that they can be successfully employed in transfer learning for molecular property predictions.

## 2 Method

### 2.1 Background

We begin this section by introducing the basic concepts which will be central in our work.

A group $G$ is a set equipped with an operation (here denoted $\cdot$) which is associative as well as having an identity element $e$ and inverse elements. In the context of data, we are mainly interested in how groups represent geometric transformations by acting on spaces and, in particular, how they describe the symmetries of an object or of a set. In either case, we are interested in how groups act on spaces. This is represented by a **group action**: given a set $X$ and a group $G$, a (left) action of $G$ on $X$ is a map $\rho : G \times X \to X$ such that it respects the group property of associativity and identity element. If $X$ is a vector space, which we will assume for the remainder of the text, we refer to group actions of the form $\rho_X : G \to \mathrm{GL}(X)$ as **representations** of $G$, where the general linear group of degree $n$ $\mathrm{GL}(X)$ is represented by the set of $n \times n$ invertible matrices. Given a group action, a concept which will play an important role in our discussion is given by the fixed points of such an action. Formally, given a point $x \in X$ and an action (representation) $\rho_X$ of $G$ on $X$, the **stabilizer** of $G$ with respect to $x$ is the subgroup $G_x = \{g \in G | \rho_X(g)x = x\} \subset G$.

In the context of representation learning, we assume our data to be defined as the space of representation-valued functions on some set $V$, i.e., $X = \{f | f : V \to W\}$. For instance, a point cloud in three dimensions can be represented as the set of functions $f : \mathbb{R}^3 \to \mathbb{Z}_2$, assigning to every point $\mathbf{r} \in \mathbb{R}^3$ the value $f(\mathbf{r}) = 0$ (the point is not included in the cloud) or $f(\mathbf{r}) = 1$ (the point is included in the cloud). Representations $\rho_V$ of a group $G$ on $V$ can be extended to representations on $f$, and therefore on $X$, $\rho_X : G \to \mathrm{GL}(X)$, as follows

$$[\rho_X(g)f](x) \equiv \rho_W(g)f(\rho_V(g^{-1})x) . \tag{1}$$

In what follows, we will then only refer to representations for the space $X$, implicitly referring to equation (1) for mapping back to how the various components transform. A map $\varphi : V \to W$ is said to be $G$-**equivariant** with respect to the actions (representations) $\rho_V, \rho_W$ if $\varphi(\rho_V(g)v) = \rho_W(g)\varphi(v)$ for every $g \in G$ and $v \in V$. Note that $G$-invariance is a particular case of the above, where we take $\rho_V, \rho_W$ to be the trivial representations. An element $x \in X$ can be described in terms of a $G$-invariant component and a group element $g \in G$, as follows: let $\varphi_{\mathrm{inv}} : X \to X/G$, be an invariant map mapping each element $x \in X$ to a corresponding canonical element $\hat{x}$ in the orbit in the quotient space $X/G$. Then for each $x \in X$ there exist a $g \in G$ such that $x = \rho_X(g)\varphi_{\mathrm{inv}}(x)$.

### 2.2 Problem Definition

We consider a classical autoencoder framework with encoding function $\eta : X \to Z$ and decoding function $\delta : Z \to X$, mapping between the data domain $X$, and latent domain $Z$, minimizing the reconstruction objective $d(\delta(\eta(x)), x)$, with a difference measure $d$ (e.g., $L_p$ norm). As discussed above, we wish to learn the invariant map $\eta$ ($\varphi_{\mathrm{inv}}$ in the previous paragraph), thus

**Property 2.1.** *The encoding function $\eta : X \to Z$ is $G$-invariant, i.e., $\eta(\rho_X(g)x) = \eta(x) \; \forall x \in X, \forall g \in G$.*

The decoding function $\delta$ maps the $G$-invariant representation $z \in Z$ back to the data domain $X$. However, as $z$ is $G$-invariant, $\delta$ can at best map $\eta(x) \in Z$ back to an element $\hat{x} \in X$ such that $\hat{x} \in \{\rho_X(g)x | \forall g \in G\}$, i.e., an element in the orbit of $x$ through $G$. This is depicted in Figure 1a. Thus, the task of the decoding function $\delta : Z \to X$ is to map encoded elements $z = \eta(x) \in Z$ to an element $\hat{x} \in X$ such that $\exists \hat{g}_x \in G$ such that

$$\delta(\eta(x)) = \hat{x} = \rho_X(\hat{g}_x)x . \tag{2}$$

We call $\hat{x}$ the **canonical** element of the decoder $\delta$. We can rewrite the reconstruction objective with a $G$-invariant encoding function $\eta$ as $d(\rho_X(\hat{g}^{-1})\delta(\eta(x)), x)$. One of the main results of this work consists in showing that $\hat{x}$ and $\hat{g}_x$ can be *simultaneously* learned by a suitable neural network. That is, we have the following property of our learning scheme:

**Property 2.2.** *There exists a **learnable** function $\psi : X \to G$ such that, given suitable $\eta, \delta$ as described above the relation $\rho_X(\psi(x))\delta(\eta(x)) = x$ , holds for all $x \in X$.*

We call any function $\psi$ satisfying (2.2) a **suitable** group function. Figure 1b describes schematically our proposed framework. In what follows, we will first characterize the defining properties of suitable group functions. Subsequently, we will describe our construction, valid for any group $G$.

## 2.3 Predicting Group Actions

In the following we further characterize the properties of $\psi$. We begin by stating two key results, while we refer to the Appendix A for the proofs.

**Proposition 2.3.** *Any suitable group function $\psi : X \to G$ is $G$-equivariant at a point $x \in X$ up the stabilizer $G_x$, i.e., $\psi(\rho_X(g)x) \subseteq g \cdot \psi(x)G_x$.*

**Proposition 2.4.** *The image of any suitable group function $\psi : X \to G$ is surjective into $\frac{G}{G_X}$, where $G_X$ is the stabilizers of all the points of $X$.*

Let us briefly discuss an example. Suppose $X = \{x = (x_0, x_1, x_2, x_3) \in \mathbb{R}^{4 \times 2} | x_i = \rho_{\mathbb{R}^2}(g_{\theta=\pi/2})^i x_0, \; x_0 \in \mathbb{R}^2\}$ and $G = \mathrm{SO}(2)$. $X$ describes all collections of vertices of squares centered at the origin of $\mathbb{R}^2$, and it is easy to check that $g_X = \mathbb{Z}_4$, generated by a $\pi/2$ rotation around the origin. In this case, any such square can be brought to any other square (of the same radius) by a rotation of an angle $\theta < \pi/2$, thus $\mathrm{Im}\psi \supseteq \{g_\theta \in \mathrm{SO}(2) | 0 \leq \theta \leq \pi/2\} = \mathrm{SO}(2)/\mathbb{Z}_4$.

Combining the two propositions above we have the following

**Lemma 2.5.** *Any suitable group function $\psi$ is an isomorphism $O_x \simeq G/G_x$ for any $x \in X$, where $O_x \subset X$ is the orbit of $x$ with respect to $G$ in $X$.*

## 2.4 Proposed Construction

Next, we turn to our proposed construction of a class of suitable group functions that satisfy Property 2.2 for any data space $X$ and group $G$. As we described above, these functions must be learnable.

**Property 2.6 (Proposed construction).** *Without loss of generality, we write our target function $\psi = \xi \circ \mu$, where $\mu : X \to Y$ is a learnable map between the data space $X$ and the embedding space $Y$, while $\xi : Y \to G$ is a deterministic map. Our construction is further determined by the following properties:*

- *We impose $\mu : X \to Y$ to be $G$-equivariant, that is, $\mu(\rho_X(g)x) = \rho_Y(g)\mu(x)$ for all $x \in X$ and $g \in G$.*

- *We ask that $Y$ is an homogeneous space, that is, given any element $y_0 \in Y$, every element $y \in Y$ can be written as $y = \rho_Y(g)y_0$ for some $g \in G$.*

- *The map $\xi : Y \to G$ is defined as follows: $\xi(y) = g$ such that $y = \rho_Y(g)y_0$ for any chosen point $y_0 \in Y$.*

In what follows we will show that our construction satisfies the properties of the previous section. For proofs see Appendix. We begin with the following

**Proposition 2.7.** *Let $\psi = \xi \circ \mu$ be a suitable group function and let $\mu : X \to Y$ be $G$-equivariant. Then, $G_x = G_{\mu(x)}$ for all $x \in X$.*

The result of the above proposition is crucial for our desired decomposition of the learned embedding, as it ensures that no information about the group action on $X$ is lost through the map $\mu$: if a group element acts non-trivially in $X$, it will also act non-trivially in $Y$.

**Proposition 2.8.** *Given $y, y_0$, the element $g$ such that $y \equiv \rho_Y(g)y_0$ is unique up to the stabilizer $G_{y_0}$.*

This proposition establishes the equivariant properties of the map $\xi$. Finally, we have

**Proposition 2.9.** *Let $\psi = \xi \circ \mu$ where $\mu$ and $\xi$ are as described above. Then, $\psi$ is a suitable group function.*

## 2.5 Intuition Behind the Proposed Framework

We conclude this rather technical section with a comment on the intuition behind our construction. Assuming for simplicity that the domain set $V$ admits the structure of vector space, $Y$ represents the space spanned by **all** basis vectors of $V$. The point $y_0$ represent a canonical orientation of such basis, and the element $\xi(y) = g$ is the group element corresponding to a basis transformation. As

all elements can be expressed in terms of coordinates with respect to a given basis, it is natural to consider a canonical basis for all orbits, justifying the assumption of homonogeneity of the space $Y$.

Further, let us assume that the invariant autoencoder correctly solves its task, $x \sim \delta(\eta(x))$. Now let $\hat{x} \in O_x$ such that $\hat{x} = \delta(\eta(x))$, and by definition, $\hat{x} = \rho_V(g)x$ for some $g \in G$. Now, the correct orbit element is identified when $\psi(\hat{x}) = e$, since $\psi(x) = g^{-1} \cdot \psi(\hat{x}) = g^{-1}$ and thus $\rho_X(g^{-1})\delta(\eta(x)) = \rho_X(g^{-1})\hat{x} = x$. Hence, during training $\psi$ needs to learn which orbit elements are decoded as "canonical", i.e., without the need of an additional group transformation. To clarify, here "canonical" does not reflect any specific property of the element, but it simply refers to the orientation learned from the decoder during training. In fact, different decoder architectures or initializations will lead to different canonical elements.

Finally, note how the different parts of our proposed framework ($\eta$, $\delta$ and $\psi$), as visualized in Figure 1b, can be jointly trained by minimizing the objective

$$d(\rho_X(\psi(x))\delta(\eta(x)), x), \tag{3}$$

which is *by construction* group invariant, i.e., not susceptible to potential group-related bias in the data (e.g., data that only occurs in certain orientations).

## 3 Application to Common Groups

In this section we describe how our framework applies to a variety of common groups which we will then implement in our experiments. As discussed in Section 2.2 and visualized in Figure 1b, the main components of our proposed framework are the encoding function $\eta$, the decoding function $\delta$ and the group function $\psi$. As stated in Property 2.1, the only constraint for the encoding function $\eta$ is that it has to be group invariant. This is in general straightforward to achieve for different groups as we will demonstrate in Section 5. Our proposed framework does not constrain the decoding function $\delta$ other than that it has to map elements from the latent space $Z$ to the data domain $X$. Hence, $\delta$ can be designed independently of the group of interest. The main challenge is in defining the group function $\psi = \xi \circ \mu$ such that it satisfies Property 2.2. Following Property 2.6 we now turn to describing our construction of $\xi$, $\mu$ and $Y$ for a variety of common groups.

**Orthogonal group $SO(2)$.** The Lie group $SO(2)$ is defined as the set of all rotations about the origin in $\mathbb{R}^2$. We take $Y$ to be the circle $S^1 \subset \mathbb{R}^2$, that is, the space spanned by unit vectors in $\mathbb{R}^2$. Now, $S^1$ is a homogeneous space: any two points $s_0, s_1 \in S^1$ are related by a rotation. Without loss of generality, we take the reference vector $y_0$ to be the vector $(1, 0) \in S^1$. Then given a vector $y \in S^1$, we can write

$$y = \begin{pmatrix} y_x \\ y_y \end{pmatrix} = \begin{pmatrix} y_x & -y_y \\ y_y & y_x \end{pmatrix} \begin{pmatrix} 1 \\ 0 \end{pmatrix}. \tag{4}$$

thus, the function $\xi : S^1 \to SO(2)$ is determined by $\xi(y) = g_\theta$ such that $\theta = \arccos(y_x) = \arcsin(y_y)$.

**Orthogonal group $SO(3)$.** We assume that $X$ has no fixed points, as this is usually the case for generic shapes (point clouds) in $\mathbb{R}^3$. It would be tempting to take $Y$ to be the sphere $S^2 \subset \mathbb{R}^3$, that is, the space spanned by unit vectors in $\mathbb{R}^3$. While this space is homogeneous, it does not satisfy the condition that the stabilizers of $G$ are trivial. In fact, given any vector $y_1 \in S^2$, we have $G_{y_1} = \{g \in SO(3) | g \text{ is a rotation about } y_1\}$.

In order to construct a space with the desired property, consider a second vector $y_2 \in S^2$ orthogonal to $y_1$, $y_2 \subset y_1^\perp$. Taking $Y$ to be the space spanned by $y_1, y_2 \in S^2$, it is easy to see that now all the stabilizers are trivial. Finally, let $y_3 = y_1 \times y_2 \in S^2$, then we construct the rotation matrix

$$R = \begin{pmatrix} y_{1,x} & y_{2,x} & y_{3,x} \\ y_{1,y} & y_{2,y} & y_{3,y} \\ y_{1,z} & y_{2,z} & y_{3,z} \end{pmatrix}, \text{ which satisfies } \begin{pmatrix} y_1 \\ y_2 \end{pmatrix} = R \begin{pmatrix} 1 & 0 & 0 \\ 0 & 1 & 0 \end{pmatrix}^\mathsf{T} = Ry_0.$$

**Symmetric group $S_n$.** A suitable space $Y$ is the set of ordered collections of unique elements of the set $M = \{1, 2, \ldots, n\}$. For instance, for $n = 3$, we have $Y =$

$\{(1,2,3),(1,3,2),(2,1,3),(2,3,1),(3,1,2),(3,2,1)\}$. It is trivial to see that the action of the permutation group on the set $Y$ is free, that is, all the stabilizers are trivial. Explicitely, given any permutation-equivariant vector $w \in \mathbb{R}^n$, we obtain an element $y = \mathrm{argsort}(w) \in Y$. Moreover, it is also obvious that any element in $Y$ can be written as $P_\sigma(1,2,\ldots,n) = (\sigma(1),\sigma(2),\ldots,\sigma(n))$, that is, a group element acting on the canonical $y_0 = (1,2,\ldots,n)$.

**Translation group $T_n$.** Here we take $Y = \mathbb{R}^n$, which is homogeneous with respect to the translation group. In fact, any vector $y \in Y$ can be trivially written as $y = y + \mathbf{0} = y + y_0$, where $\mathbf{0}$ is the origin of $\mathbb{R}^n$. Our group function takes therefore the form $\xi(y) = y$.

**Euclidean group SE$(n)$.** A generic transformation of the Euclidean group on a $n$-dimensional representation $v \in V$ is

$$v \mapsto Av + b \,, \quad A \in \mathrm{SO}(n)\,, b \in T_n \,. \tag{5}$$

Let $\mu = (\mu_1, \mu_2, \ldots, \mu_{n+1})$ be a collection of $n+1$ $n$-dimensional SE$(n)$-equivariant vectors, that is, $\mu_i(\rho_X(g)x) = \rho_Y(g)\mu(x)$, $i = 1,\ldots,n$. We construct $\widehat{y}_a = (\mu_a - \mu_{n+1})/\|\mu_a - \mu_{n+1}\| \in S^n$, $a = 1,\ldots,n$, where $S^n$ is the unit $n$-dimensional sphere. These $n$ ortho-normal vectors are **translation invariant** but rotation equivariant, and are suitable to construct the rotation matrix

$$R = (\widehat{y}_1 \quad \widehat{y}_2 \quad \cdots \quad \widehat{y}_n) \,, \tag{6}$$

while the extra vector $\widehat{y}_{n+1} = \mu_{n+1}$ can be used to predict the translation action. Putting all together, the space $Y$ is described by $n$ vectors $y_a = \widehat{y}_a + \widehat{y}_{n+1}$, and $y_0 = I_n$ is the $n \times n$ unit matrix, as

$$(R + \widehat{y}_{n+1})I_n = (\widehat{y}_1 \quad \cdots \quad \widehat{y}_n)^\mathsf{T} + \widehat{y}_{n+1}I_n = (y_1 \quad \cdots \quad y_n)^\mathsf{T} \,. \tag{7}$$

# 4 Related Work

**Group equivariant neural networks.** Group equivariant neural networks have shown great success for various groups and data types. There are two main approaches to implement equivariance in a layer and, hence, in a neural network. The first, and perhaps the most common, imposes equivariance on the space of functions and features learned by the network. Thus, the parameters of the model are constrained to satisfy equivariance Thomas et al. (2018); Weiler & Cesa (2019a); Weiler et al. (2018a); Esteves et al. (2020). The disadvantage of this approach consists in the difficulty of designing suitable architectures for all components of the model, transforming correctly under the group action Xu et al. (2021). The second approach to equivariance consists in lifting the map from the space of features to the group $G$, and equivariance is defined on functions on the group itself Romero & Hoogendoorn (2020); Romero et al. (2020); Hoogeboom et al. (2018). Although this strategy avoids the architectural constraints, applicability is limited to homogeneous spaces Hutchinson et al. (2021) and involves an increased dimensionality of the feature space, due to the lifting to $G$. Equivariance has been explored in a variety of architecture and data structures: Convolutional Neural Networks Cohen & Welling (2016a); Worrall et al. (2017); Weiler et al. (2018c); Bekkers et al. (2018); Thomas et al. (2018); Dieleman et al. (2016); Kondor & Trivedi (2018); Cohen & Welling (2016b); Cohen et al. (2018); Finzi et al. (2020), Transformers Vaswani et al. (2017); Fuchs et al. (2020); Hutchinson et al. (2021); Romero & Cordonnier (2020), Graph Neural Networks Defferrard et al. (2016); Bruna et al. (2013); Kipf & Welling (2016); Gilmer et al. (2017); Satorras et al. (2021) and Normalizing Flows Rezende & Mohamed (2015); Köhler et al. (2019, 2020); Boyda et al. (2021). These methods are usually trained in a supervised manner and combined with a symmetric function (e.g. pooling) to extract group-invariant representations.

**Group equivariant autoencoders.** Another line of related work is concerned with group equivariant autoencoders. Such models utilize specific network architectures to encode and decode data in an equivariant way, resulting into equivariant representations only Hinton et al. (2011); Sabour et al. (2017); Kosiorek et al. (2019); Guo et al. (2019). Feige (2019) use weak supervision in an AE to extract invariant and equivariant representations. Winter et al. (2021) implement a permutation-invariant AE to learn graph embeddings, in which the permutation matrix for graph matching is learned during training. In that sense, the present work can be seen as a generalization of their approach for a generic data type and any group.

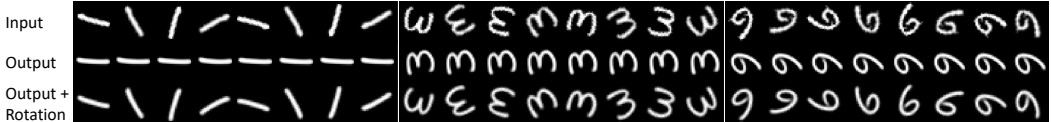

Figure 3: Input and predicted output for rotated versions of three MNIST images. Top row shows the input image successively rotated by $45°$. Middle row shows the decoded (canonical) image and bottom row shows the decoded image after applying the predicted rotation.

**Unsupervised invariant representation learning.** The field of unsupervised invariant representation learning can be roughly divided into two categories. The first consists in learning an approximate group action in order to match the input and the reconstructed data. For instance, Mehr et al. (2018b) propose to encode the input in quotient space, and train the model with a loss that is defined by taking the infimum over the group $G$. While this is feasible for (small) finite groups, for continuous groups they either have to approximately discretize them or perform a separate optimization of the group action at every back propagation step to find the best match. Other work Shu et al. (2018); Koneripalli et al. (2020) proposes to disentangle the embedding in a shape-like and a deformation-like component. While this is in spirit with our work, their transformations are local (we focus on global transformations) and are approximative, that is, the components are not explicitly invariant and equivariant with respect to the transformation, respectively.

In the case of 2D/3D data, co-alignment of shapes can be used to match the input and the reconstructed shapes. Some approaches are unfeasible Wang et al. (2012) as they are not compatible with a purely unsupervised approach, while other Averkiou et al. (2016); Chaouch & Verroust-Blondet (2008, 2009) leverage symmetry properties of the data and PCA decomposition, exhibiting however limitation regarding scalability. For graphs, the problem of graph matching Bunke & Jiang (2000) has been tackled in several works and with different approaches, for instance algorithmically, e.g., Ding et al. (2020), or by means of a GNN Li et al. (2019).

On the topic of group theory-based embedding disentanglement, some works are based on the definition of Higgins et al. (2018) of a disentangled representations. We refer to this as "symmetry-based decomposition", where the various factors in the disentangled representation correspond to the decomposition of symmetry groups acting on the data space. In Pfau et al. (2020), the authors show that, with some assumption on the geometry of the underlying data space, it is possible to learn to factorize a Lie group from the orbits in data space. The works Hosoya (2019); Keurti et al. (2022), for instance, design unsupervised generative VAEs approaches for learning representation corresponding to orthogonal symmetry actions on the data space. In our work, on the other hand, we learn a decomposition into separate *group representations*. These are all representations of the same group, but act differently on different data space (analogously to different SO(3) representations identified by the angular quantum number $l = 0, 1, 2, \dots$).

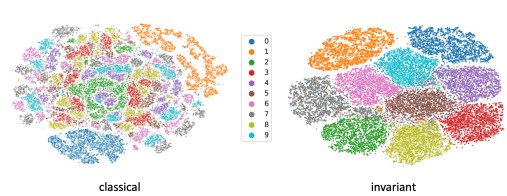

Figure 2: TSNE embedding of the encoded test dataset for a classical and our proposed SO(2) invariant autoencoder.

## 5 Experiments

In this section we present differnt experiments for the various groups discussed in Section 3. [2]

### 5.1 Rotated MNIST

In the first experiment, we train an SO(2)-invariant autoencoder on the original (non-rotated) MNIST dataset and validate the trained model on the rotated MNIST dataset (ref. mni) which consists of randomly rotated versions of the original MNIST dataset. For the functions $\eta$ and $\psi$ we utilize SO(2)-Steerable Convolutional Neural Networks Weiler & Cesa (2019b). For more details about the

---

[2]Source code for the different implementations available at `https://github.com/jrwnter/giae`.

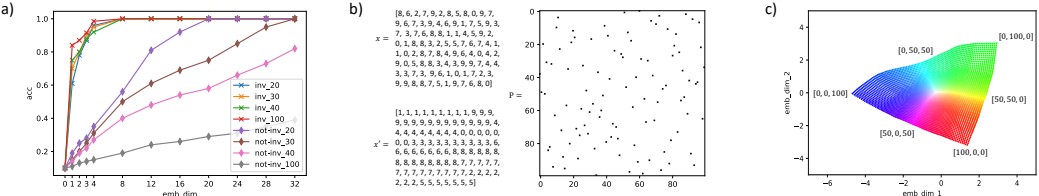

Figure 4: a) Element-wise reconstruction accuracy of our proposed permutation invariant autoencoder (cross) and a classical non-permutation invariant autoencoder (diamond) for different embedding and set sizes. b) Example set $x$ with 100 elements with its canonical reconstruction $\hat{x}$ and the predicted permutation matrix $P$ (resulting into a perfect reconstruction). One can confirm for oneself that, e.g., $x[38] = x'[0]$, matching $P[38, 0] = 1$. c) Best viewed in colour. Visualization of the two-dimensional embedding of a permutation-invariant autoencoder for all 5151 sets of 100 elements with 3 different element classes. Each point represents one set, colours represent set compositions (proportion of each element class, independent of the order).

network architecture and training, we refer to Appendix B. In Figure 3 we show images in different rotations and the respective reconstructed images by the trained model. The model decodes the different rotated versions of the same image (i.e., elements from the same orbit) to the same canonical output orientation (second row in Figure 3). The trained model manages to predict the right rotation matrix (group action) to align the decoded image with the input image, resulting in an overall low reconstruction error. Note that the model never saw rotated images during training but still manages to encode and reconstruct them due to its inherent equivariant design. We find that the encoded latent representation is indeed rotation invariant (up to machine precision), but only for rotations of an angle $\theta = \frac{n \cdot \pi}{2}$, $n \in \mathbb{N}$. For all other rotations, we see slight variations in the latent code, which, however, is to be expected due to interpolation artifacts for rotations on a discretized grid. Still, inspecting the 2d-projection of the latent code of our proposed model in Figure 2, we see distinct clusters for each digit class for the different images from the test dataset, independent of the orientation of the digits in the images. In contrast, the latent code of a classical autoencoder exhibits multiple clusters for different orientations of the same digit class.

## 5.2 Set of Digits

Next, we train a permutation-invariant autoencoder on sets of digits. A set with $N$ digits is represented by concatenating one-hot vectors of each digit in a $N \times D$-dimensional matrix, where we take $D = 10$. Notice that this matrix-representation of a set is *not* permutation invariant. We randomly sampled 1.000.000 different sets for training and 100.000 for the final evaluation with $N = 20, 30, 40, 100$, respectively, removing all permutation equivariant sets (i.e., there are no two sets that are the same up to a permutation). For comparison, we additionally trained a classical non-permutation-invariant autoencoder with the same number of parameters and layers as our permutation-invariant version. For more details on the network architecture and training we refer to Appendix C. Here, we demonstrate how the separation of the permutation-invariant information of the set (i.e., the composition of the set) from the (irrelevant) order-information results in a significant reduction of the space needed to encode the set. In Figure 4a, we plot the element-wise reconstruction accuracy of different sized sets for both models for varying embedding (bottleneck) sizes. As the classical autoencoder has to store both the composition of digits in the set (i.e., number of elements for each of the 10 digits classes) as well as their order in the permutation-dependent matrix representation, the reconstruction accuracy drops for increasing size of the set $N$ for a fixed embedding size. For the same reason, perfect reconstruction accuracy is only achieved if the embedding dimension is at least as large as the number of digits in the set. On the contrary, our proposed permutation invariant autoencoder achieves perfect reconstruction accuracy with a significant lower embedding size. Crucially, as no order information has to be stored in the embedding, this embedding size for perfect reconstruction accuracy also stays the same for increasing size $N$ of the set. In Figure 4b we show one example for a set $x$ with $N = 100$ digits, with the predicted canonical orbit element $\hat{x}$ and the predicted permutation matrix. As perhaps expected, the canonical element clusters together digits with same value, while not using the commonly used order of Arabic numerals. This learned order (here [1,9,4,0,3,6,8,7,2,5]) stays fixed for the trained network for different inputs but changes upon re-initialization of the network.

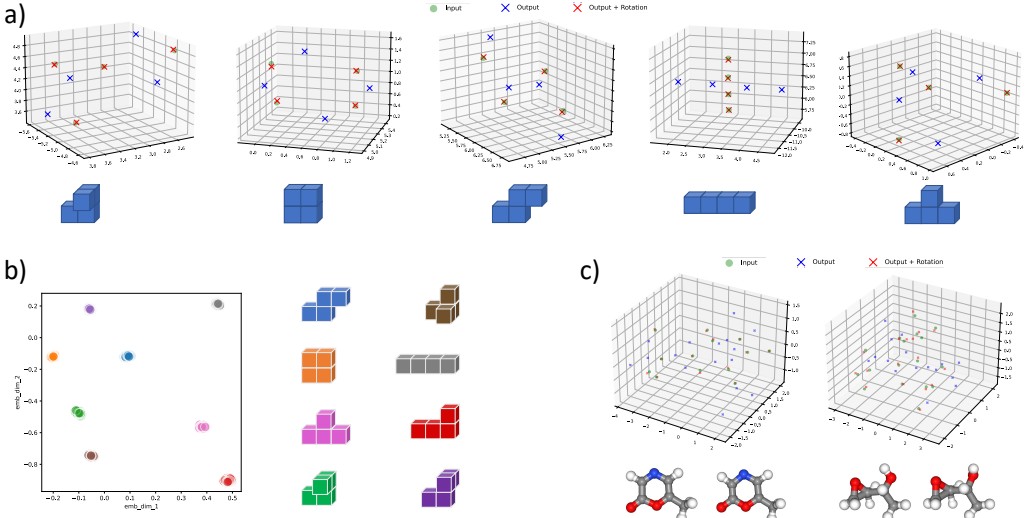

Figure 5: a) Five different Tetris shapes represented by points at the center of the four blocks respectively. Input points, output points and rotated (predicted group action) output points as reconstructed by our proposed SE(3)- and S($N$)-invariant autoencoder are visualized. b) Two-dimensional latent space for all Tetris shapes augmented with Gaussian noise ($\sigma = 0.01$). Colors of points match colors of shapes on the right. c) Two molecular conformations and their reconstructions represented as point cloud and ball-and-stick model (left true, right predicted).

In Figure 4c we show the two-dimensional embedding of a permutation invariant autoencoder trained on set of $N = 100$ elements chosen from $D = 3$ different classes (e.g. digits 0,1,2). As the sets only consists of 3 different elements (but in different compositions and order) we can visualize the $\binom{D+N-1}{N} = \binom{102}{100} = 5151$ elements in the two-dimensional embedding and colour them according to their composition. As our proposed autoencoder only needs to store the information about the set composition and not the order, the embedding is perfectly structured with respect to the composition as can be seen by the colour gradients in the visualization of the embedding.

## 5.3 Point Cloud

Point clouds are a common way to describe objects in 3D space, such as the atom positions of a molecule or the surface of an object. As such, they usually adhere to 3D translation and rotation symmetries and are unordered, i.e., permutation invariant. Hence, we investigate in the next experiment a combined SE(3)- and $S_N$-invariant autoencoder for point cloud data. We use the Tetris Shape toy dataset Thomas et al. (2018) which consists of 8 shapes, where each shape includes $N = 4$ points in 3D space, representing the center of each Tetris block. To generate various shapes, we augment the 8 shapes by adding Gaussian noise with $\sigma = 0.01$ standard deviation on each node's position. Different orientations are obtained by rotating the point cloud with a random rotation matrix $R \in$ SO(3) and further translating all node positions with the same random translation vector $t \in \mathbb{R}^3 \simeq T_3$. For additional details on the network architecture and training we refer to Appendix D. In Figure 5 we visualize the input points and output points before and after applying the predicted rotation. The model successfully reconstructs the input points with high fidelity (mean squared error of $\sim 4 \times 10^{-5}$) for all shapes and arbitrary translations and rotations. Figure 5b shows the two-dimensional embedding of the trained SE(3)- and $S_N$-invariant autoencoder. Augmenting the points with random noise results into slight variations in the embedding, while samples of the same Tetris shape class still cluster together. The embedding is invariant with respect to rotations, translation and permutations of the points. Notably, the SE(3)-invariant representations can distinguish the two chiral shapes (compare green and violet coloured shapes in the bottom right of Figure 5b). These two shapes are mirrored versions of themselves and should be distinguished in an SE(3) equivariant model. Models that achieve SE(3) invariant representations by restricting themselves to composition of symmetric functions only, such as working solely on distances (e.g. SchNet Schütt et al. (2018)) or angles (e.g. ANI-1 Smith et al. (2017)) between points fail to distinguish these two shapes Thomas et al. (2018).

**Molecular Conformations.** We showcase our learning framework on real-world data by autoencoding the atom types and geometries of small molecules from the QM9 database Ramakrishnan et al. (2014). We achieved a reconstruction RMSE of $0.15 \pm 0.07$ Å for atom coordinates and perfect atom type accuracy on 5000 unseen test conformations (see Figure 5c for two examples and Appendix E.2.2 for more reconstruction predictions). Given a point cloud of $N$ nodes, the $G = SE(3) \times S_N$-invariant embedding $z$ has to store information about the Cartesian coordinates $P \in \mathbb{R}^{3N}$ as well as the 5 distinct atom types $A \in \{0,1\}^{5N}$ represented as one-hot encodings. The largest molecule in the QM9 database has $N_{\max} = 29$ atoms, thus the degrees of freedom of the data space[3] $X$ are $3 \cdot 29 \cdot 5 \cdot 29 = 12615$. Our embeddings compress this high-dimensional space of molecular conformations into $z \in Z \subset \mathbb{R}^{256}$ dimensions.

## 5.4 ShapeNet

We also run experiments on the ShapeNet dataset Chang et al. (2015). We utilized 3D Steerable CNNs proposed by Weiler et al. (2018b) as equivariant encoder for the 3d voxel input space. We utilized the scalar outputs as rotation-invariant embedding ($z$) and predict (analogously to our experiments on 3d point clouds) 2 rotation-equivariant vectors to construct a rotation matrix $\rho(g)$. In Figure 11 in the

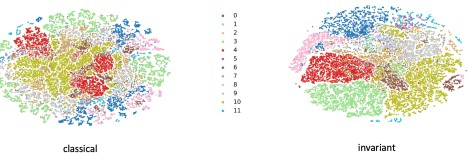

Figure 6: TSNE embedding of the encoded dataset for a classical autoencoder and our proposed SO(3) invariant autoencoder.

Appendix we show example reconstructions of shapes from the SE(3) invariant representations. Similar to our MNIST experiment, we compared the resulting embedding space to the embeddings produced by a non-invariant autoencoder model. As the dataset comes in an aligned form (e.g., cars are always aligned in the same orientation), we additionally applied random 90 degree rotations to remove this bias (while avoiding interpolation artifacts) when training the non-invariant model. Random rotations are also applied to the common test set. In Figure 6 we visualize a TSNE projection of the embeddings of both models. We can see a well structured embedding space for our model with distinct clusters for the different shape classes. On the other hand, the embeddings produced by the non-invariant autoencoder is less structured and one can make out different clusters for the same shape label but in different orientations. Moreover, we compared the downstream performance and generalizability of a KNN classifier on shape classification, trained on 1000 embeddings and tested on the rest. The classifier based on our rotation-invariant embeddings achieved an accuracy of 0.81 while the classifier based on the non-invariant embeddings achieved an accuracy of only 0.63.

## 6 Conclusion and Future Work

In this work we proposed a novel unsupervised learning strategy to extract representations from data that are separated in a group invariant and equivariant part for any group $G$. We defined the sufficient conditions for the different parts of our proposed framework, namely the encoder, decoder and group function without further constraining the choice of a ($G$-) specific network architecture. In fact, we demonstrate the validity and flexibility of our proposed framework for diverse data types, groups and network architectures.

To the best of our knowledge, we propose the first general framework for unsupervised learning of separated invariant-equivariant representations valid for any group. Our learning strategy can be applied to any AE framework, including variational AEs. It would be compelling to extend our approach to a fully probabilistic approach, where the group action function samples from a probability distribution. Such formalism would be relevant in scenarios where some elements of a group orbit occur with different frequencies, enabling this to be reflected in the generation process. For instance, predicting protein-ligand binding sites depends on the molecule's orientation with respect to the protein pocket or cavity. Thus, in a generative approach, it would be highly compelling to generate a group action reflecting a candidate molecule's orientation in addition to a candidate ligand. We plan to return to these generalization and apply our learning strategy to non-trivial real-world applications in future work.

---

[3]Notice that the data space $X$ can be described as the product space between $\mathbb{R}^{3N}$ and $\mathbb{N}^{5N}$.

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
