# Appendix

## A  Proofs

**Proposition A.1.** *Any suitable group function $\psi : X \to G$ is G-equivariant at a point $x \in X$ up the stabilizer $G_x$, i.e., $\psi(\rho_X(g)x) \subseteq g \cdot \psi(x)G_x$.*

*Proof*: As the relation (see Property 2.2)

$$\rho_X(\psi(x))\delta(\eta(x)) = x \tag{8}$$

must hold for any $x \in X$, it must hold for any point $x' = \rho_X(g)x$ in the orbit of $x$, which then reads

$$\begin{aligned} x' &= \rho_X(\psi(x'))\delta(\eta(x')) \\ &= \rho_X(\psi(\rho_X(g)x))\delta(\eta(x)) \ , \end{aligned} \tag{9}$$

where we used the invariance of $\eta$. On the other hand, applying $\rho_X(g)$ to both sides of (8) we have

$$\begin{aligned} \rho_X(g)x &= \rho_X(g)\rho_X(\psi(x))\delta(\eta(x)) \\ &= \rho_X(g\psi(x))\delta(\eta(x)) \ , \end{aligned} \tag{10}$$

since $\eta(\rho_X(g')x) = \eta(x)$ and $\rho_X(g_1)\rho_X(g_2) = \rho_X(g_1 g_2)$. Combining (9) and (10) it follows that

$$\rho_X(\psi(x)^{-1} \cdot g^{-1} \cdot \psi(\rho_X(g)x))\delta(\eta(x)) = \delta(\eta(x)) \ , \tag{11}$$

that is, $\psi(x)^{-1} \cdot g^{-1} \cdot \psi(\rho_X(g)x)) \in G_{\delta(\eta(x))}$. Now, since $x$ and $\delta(\eta(x))$ by assumption belong to the same orbit of $G$, it follows that they have isomorphic stabilizers, $G_{\delta(\eta(x))} \simeq G_x$. Thus, we have shown that $\psi(\rho_X(g)x)) = g \cdot \psi(x) \cdot g'$, where $g' \in G_x$, which proves our claim. $\square$

**Proposition A.2.** *The image of any suitable group function $\psi : X \to G$ is surjective into $\frac{G}{G_X}$, where $G_X$ is the stabilizers of all the points of $X$.*

*Proof*: Let $x \in X$ be such that $x = \delta(\eta(x))$, that is, $\psi(x) = G_x$, the stabilizer of $x$. Note that each orbit cointains at least one such element. For any element $g \in G$ we have that, using Proposition A.1, $\psi(\rho_X(g)x) = g \cdot \psi(x) \cdot \tilde{g}$, where $\tilde{g} \in G_x$. Since $\psi(x) \cdot \tilde{g} \in G_x$ as well, it then follows that the image of $\psi$ is $G$ up to an action by an element of the stabilizer $G_x$. Applying the above reasoning to every points $x \in X$, we have that $\text{Im}(\psi) = \cup_{x \in X}\frac{G}{G_x} = \frac{G}{\cap_{x \in X}G_x}$, where $\cap_{x \in X}G_x = G_X = \{g \in G | \rho_X(g)x = x, \ \forall x \in X\}$, proving our claim.$\square$

**Lemma A.3.** *Any suitable group function $\psi$ is an isomorphism $O_x \simeq G/G_x$ for any $x \in X$, where $O_x \subset X$ is the orbit of $x$ with respect to $G$ in $X$.*

*Proof*: Surjectivity follows directly from Proposition A.2. To show injectivity, consider $x, x' \in O_x$ such that $\psi(x') = \psi(x) \cdot \tilde{g}$, where $\tilde{g} \in G_x$. From Proposition 2.3 it follows that $x' = x$, which proves the claim. $\square$

**Proposition A.4.** *Let $\psi = \xi \circ \mu$ be a suitable group function and let $\mu : X \to Y$ be G-equivariant. Then, $G_x = G_{\mu(x)}$ for all $x \in X$.*

*Proof*: Let $g \in G_x$, that is, $\rho_X(g)x = x$. Applying $\mu$ to both sides of this equation we obtain $\mu(x) = \mu(\rho_X(g)x) = \rho_Y(g)\mu(x)$, where we used the G-equivariance of $\mu$. Hence, $G_x \subseteq G_{\mu(x)}$. To prove the opposite inclusion, let $g \in G_{\mu(x)}$ but $g \notin G_x$, and let $x' = \rho_X(g)x$. Now, $\mu(x') = \rho_Y(g)\mu(x) = \mu(x)$, thus $\mu$, and therefore $\psi = \xi \circ \mu$, maps the distinct element $x, x'$ to the same group element $\psi(x) = \psi(x')$, in contradiction with Proposition 2.3. $\square$

**Proposition A.5.** *Given $y, y_0$, the element $g$ such that $y \equiv \rho_Y(g)y_0$ is unique up to the stabilizer $G_{y_0}$.*

*Proof*: Suppose that there exist $g_1, g_2 \in G$ such that $\rho_Y(g_1)y_0 = \rho_Y(g_2)y_0$, then $\rho_Y(g_2^{-1}g_1)y_0 = y_0$, which implies $g_2^{-1}g_1 \in G_{y_0}$. $\square$

**Proposition A.6.** *Let $\psi = \xi \circ \mu$ where $\mu$ and $\xi$ are as described above. Then, $\psi$ is a suitable group function.*

*Proof*: We show that our construction describes an isomorphism $O_x \simeq G/G_x$ for all $x \in X$. Given $x \in X$ and $g \in G$, Propositions A.4 and A.5 imply

$$\xi\left(\mu(\rho_X(g)x)\right) = \xi(\rho_Y(g)\mu(x)) \subseteq g \cdot \xi(\mu(g))G_x , \tag{12}$$

that is, $\psi$ possesses the $G$-equivariant property as required in Proposition 2.3, which in turns imply injectivity, as in Lemma A.3. Surjectivity follows from the same argument as in Proposition A.2, since the proof only relies on the equivariant properties of $\psi$, which we showed in (12). □

## B   Model architecture Rotated MNIST

We follow Weiler & Cesa (2019b) and use steerable CNNs to parameterize functions $\eta$ and $\mu$. In contrast to classical CNNs, CNNs with O(2)-steerable kernels transform feature fields respecting the transformation law under actions of $O(2)$. We can define scalar fields $s : \mathbb{R}^2 \to \mathbb{R}$ and vector fields $v : \mathbb{R}^2 \to \mathbb{R}^2$ that transform under group actions (rotations) the following:

$$s(x) \mapsto s(g^{-1}x) \qquad v(x) \mapsto g \cdot v(g^{-1}x) \qquad \forall g \in O(2) . \tag{13}$$

Thus, scalar values are moved from one point on the plane $\mathbb{R}^2$ to another but are not changed, while vectors are moved and changed (rotated) equivalently. Hence, we can utilize steerable CNNs to encode samples in $\mathbb{R}^2$ in $O(2)$-invariant scalar features and $O(2)$-equivariant vector features. We can use the scalar features $s$ as $\mathcal{G}$-invariant representation $z \in Z$ and following Section 3 (Orthogonal group SO(2)) utilizing a single vector features $v$ to construct the rotation matrix $R$ as:

$$R = \begin{bmatrix} \bar{v}_x & -\bar{v}_y \\ \bar{v}_y & \bar{v}_x \end{bmatrix}, \qquad \bar{v} = \frac{v}{\|v\|}. \tag{14}$$

In our experiments we used seven layers of steerable CNNs as implemented by Weiler & Cesa (2019b). We did not use pooling layers, as we found them to break rotation equivariance and only averaged over the two spatial dimensions after the final layer to extract the final invariant embedding and equivariant vector. In each layer we used 32 hidden scalar and 32 hidden vector fields. In the final layer we used 32 scalar fields (32 dimensional invariant embedding) and one vector feature field.

The Decoding function $\delta : Z \to \mathbb{R}^2$ can be parameterized by a regular CNN. In our experiments we used six layers of regular CNNs with 32 hidden channels, interleaved with bilinear upsampling layers starting from the embedding expanded to a $2 \times 2 \times 32$ tensor.

Training was done on one NVIDIA Tesla V100 GPU in approximately 6 hours.

## C   Model architecture Set of Digits

We can rewrite the equation $P_\sigma(1, 2, \ldots, n) = (\sigma(1), \sigma(2), \ldots, \sigma(n))$, in vector form by representing set elements by standard $n \times 1$ column vectors $\mathbf{e}_i$ (one-hot encoding) and $\sigma$ by a permutation matrix $P_\sigma$ whose (i,j) entry is 1 if $i = \sigma(j)$ and 0 otherwise, then:

$$P_\sigma \mathbf{e}_i = \mathbf{e}_{\sigma(i)} \tag{15}$$

Hence, encoding function $\eta$ should encode a set of elements in a permutation invariant way and $\psi$ should map a set $M$ to a permutation matrix $P_\sigma$:

$$\psi : M \to P_\sigma \tag{16}$$

We follow Zaheer et al. (2017) and parameterize $\eta$ by a neural network $\gamma$ that is applied element-wise on the set followed by an invariant aggregation function $\Sigma$ (e.g. sum or average) and a second neural network $\beta$:

$$\eta(X) = \beta(\Sigma_{x \in X} \gamma(x)) . \tag{17}$$

In our experiments we parameterized $\gamma$ and $\beta$ with regular feed-forward neural networks with three layers respectively, also using ReLU activations and Batchnorm.

The output of function $\gamma$ is equivariant and can also be used to construct $\psi$. We follow Winter et al. (2021) and define a function $s : \mathbb{R}^d \to \mathbb{R}$ mapping the output of $\gamma$ for every set element to a scalar

value. By sorting the resulting scalars, we construct the permutation matrix $P_\sigma$ with entries $p_{ij}$ that would sort the set of elements with respect to the output of $s$:

$$p_{ij} = \begin{cases} 1, & \text{if } j = \operatorname{argsort}(s)_i \\ 0, & \text{else} \end{cases} \tag{18}$$

As the argsort operation is not differentiable, we utilizes a continuous relaxation of the argsort operator proposed in (Prillo & Eisenschlos, 2020; Grover et al., 2019):

$$\mathbf{P} \approx \hat{\mathbf{P}} = \operatorname{softmax}\left(\frac{-d(\operatorname{sort}(s)1^\top, 1s^\top)}{\tau}\right), \tag{19}$$

where the softmax operator is applied row-wise, $d(x, y)$ is the $L_1$-norm and $\tau \in \mathbb{R}_+$ a temperature-parameter.

Decoding function $\delta$ can be parameterized by a neural network that maps the permutation-invariant set representation back to either the whole set or single set elements. In the letter case, where the same function is used to map the same set representation to the different elements, additional fixed position embeddings can be fed into the function to decode individual elements for each position/index. For the reported results we choose this approach, using one-hot vectors as position embeddings and a 4-layer feed-forward neural network.

Training was done on one NVIDIA Tesla V100 GPU in approximately 1 hours.

# D    Model architecture Point Cloud - Tetris 3D & QM9

We implement a graph neural network (GNN) that transform equivariantly under rotations and translations in 3D space, respecting the invariance and equivariance constraints mentioned in Eq. (6) and (7) for $n = 3$.

Assume we have a point cloud of $N$ particles each located at a certain position $x_i \in \mathbb{R}^3$ in Cartesian space. Now given some arbitrary ordering $\sigma(\cdot)$ for the points, we can store the positional coordinates in the matrix $P = [x_1, ..., x_N] \in \mathbb{R}^{N \times 3}$. Standard Graph Neural Networks (GNNs) perform message passing Gilmer et al. (2017) on a local neighbourhood for each node. Since we deal with a point cloud, common choice is to construct neighbourhoods through a distance cutoff $c > 0$. The edges of our graph are specified by *relative* positions

$$x_{ij} = x_j - x_i \in \mathbb{R}^3 \,,$$

and the neighbourhood of node $i$ is defined as $\mathcal{N}(i) = \{j : d_{ij} := ||x_{ij}|| \leq c\}$.

Now, our data (i.e., the point cloud) lives on a vector space $X$, where we want to learn an SE(3) invariant and equivariant embedding wrt. arbitrary rotations and translations in 3D space. Let the feature for node $i$ consist of an invariant (type-0) embedding $h_i \in \mathbb{R}^{F_s}$, an equivariant (type-1) embedding $w_i \in \mathbb{R}^{3 \times F_v}$ that transforms equivariantly wrt. arbitrary rotation **but** is invariant to translation. Such a property can be easily obtained, when operating with relative positions.

Optionally, we can model another equivariant (type-1) embedding $t_i \in \mathbb{R}^3$ which transforms equivariantly wrt. translation **and** rotation. As our model needs to learn to predict group actions in the SE(3) symmetry, we require to predict an equivariant translation vector ($b \in T_3$), as well as a rotation matrix ($A \in$ SO(3)), where we will dedicate the $t$ vector to the translation and the $w$ vector(s) to the rotation matrix.

As point clouds might not have initial features, we initialize the SE(3)-invariant embeddings as one-hot encoding $h_i = \mathbf{e}_i$ for each node $i = 1, \ldots, N$. The (vector) embedding dedicated for predicting the rotation matrix is initialized as zero-tensor for each particle, i.e., $w_i = \mathbf{0}$ and the translation vector is initialized as the absolute positional coordinate, i.e. to, $t_i = x_i$.

We implement following edge function $\phi_e : \mathbb{R}^{2F_s+1} \mapsto \mathbb{R}^{F_s + 2F_v + k}$ with

$$m_{ij} = \phi_e(h_i, h_j, d_{ij}) = W_e[h_i, h_j, d_{ij}] + b_e, \tag{20}$$

and set $k = 1$ if the GNN should model the translation and $k = 0$ else. Notice that the message $m_{ij}$ in Eq. (20) only depends on SE(3) invariant embeddings. Now, (assuming $k = 1$) we further split the message tensor into 4 tensors,

$$m_{ij} = [m_{h,ij}, m_{w_0,ij}, m_{w_1,ij}, m_{t,ij}] \,,$$

which we require to compute the aggregated messages for the SE(3) invariant and equivariant node embeddings.

We include a row-wise transform $\phi_s : \mathbb{R}^{F_s} \mapsto \mathbb{R}^{F_s}$ for the invariant embeddings using a linear layer:

$$\tilde{h}_i = W_s h_i + b_s \,, \tag{21}$$

The aggregated messages for invariant (type-0) embedding $h_i$ are calculated using:

$$m_{i,h} = \sum_{j \in \mathcal{N}(i)} m_{h,ij} \odot \tilde{h}_i \quad \in \mathbb{R}^{F_s}. \tag{22}$$

where $\odot$ is the (componentwise) scalar-product.

The aggregated equivariant features are computed using the tensor-product $\otimes$ and scalar-product $\odot$ from (invariant) type-0 representations with (equivariant) type-1 representations:

$$m_{i,w} = \sum_{j \in \mathcal{N}(i)} \left( x_{ij} \otimes m_{w_0,ij} + (w_i \times w_j) \odot (\mathbf{1} \otimes m_{w_1,ij}) \right) \quad \in \mathbb{R}^{3 \times F_v} \,, \tag{23}$$

where $\mathbf{1} \in \mathbb{R}^3$ is the vector with 1's as components and $(a \times b)$ denotes the cross product between two vectors $a, b \in \mathbb{R}^3$.

The tensor in Eq. (23) is equivariant to arbitary rotations and invariant to translations. It is easy to prove the translation invariance, as any translation $t^* \in T_3$ acting on points $x_i, x_j$ does not change the relative position $x_{ij} = (x_j + t^*) - (x_i + t^*) = x_j - x_i$.

To prove the rotation equivariance, we first observe that given any rotation matrix $A \in \mathrm{SO}(3)$ acting on the provided data, as a consequence relative positions rotate accordingly, since

$$A x_j - A x_i = A(x_j - x_i) = A x_{ji} \in \mathbb{R}^3.$$

The tensor product $\otimes$ between two vectors $u \in \mathbb{R}^3$ and $v \in \mathbb{R}^{F_s}$, commonly also referred to as *outer product* is defined as

$$u \otimes v = u v^\top \in \mathbb{R}^{3 \times F_s} \,,$$

and returns a matrix given two vectors. For the case that a group representation of $\mathrm{SO}(3)$, i.e. a rotation matrix $R$, acts on $u$, it is obvious to see with the associativity property

$$(Au) \otimes v = (Au)v^\top = Auv^\top = A(uv^\top) = A(u \otimes v) = Au \otimes v.$$

The cross product $(w_i \times w_j) \in \mathbb{R}^{3 \times F_v}$ used in equation (23) between type-1 features $w_i$ and $w_j$ is applied separately on the last axis. The cross product has the algebraic property of rotation invariance, i.e. given a rotation matrix $A$ acting on two 3-dimensional vectors $a, b \in \mathbb{R}^3$ the following holds:

$$(Aa) \times (Ab) = A(a \times b) \,. \tag{24}$$

Now, notice that the quantities that "transform as a vector" which we call type-1 embeddings are in $S = \{x_{ij}, w_i, t_i\}_{i,j=1}^N$.

Given a rotation matrix $A$ acting on elements of $S$, we can see that the result in (23)

$$\sum_{j \in \mathcal{N}(i)} \left( A x_{ij} \otimes m_{w_0,ij} + (A w_i) \times (A w_j) \odot (\mathbf{1} \otimes m_{w_1,ij}) \right)$$

$$= \sum_{j \in \mathcal{N}(i)} \left( A x_{ij} \otimes m_{w_0,ij} + A(w_i \times w_j) \odot (\mathbf{1} \otimes m_{w_1,ij}) \right)$$

$$= A \sum_{j \in \mathcal{N}(i)} \left( x_{ij} \otimes m_{w_0,ij} + (w_i \times w_j) \odot (\mathbf{1} \otimes m_{w_1,ij}) \right)$$

$$= A m_{i,w}$$

is rotationally equivariant.

We update the hidden embedding with a residual connection

$$\begin{aligned} h_i &\leftarrow h_i + m_{i,h} \,, \\ w_i &\leftarrow w_i + m_{i,w} \,, \end{aligned} \tag{25}$$

and use a Gated-Equivariant layer with equivariant non-linearities as proposed in the PaiNN architecture [Schütt et al. (2021)] to enable an information flow between type-0 and type-1 embeddings. The type-1 embedding for the translation vector is updated in a residual fashion

$$t_i \leftarrow t_i + \sum_{j \in \mathcal{N}(i)} x_{ij} \otimes m_{t,ij}$$

$$= t_i + \sum_{j \in \mathcal{N}(i)} m_{t,ij} \odot x_{ij}, \tag{26}$$

where we can replace the tensor-product with a scalar-product, as $m_{t,ij} \in \mathbb{R}$. The result in Eq. (26) is translation and rotation equivariant as the first summand $t_i$ is rotation and translation equivariant, while the second summand is only rotation equivariant since we utilize relative positions.

For the SE(3) Tetris experiment, the encoding function $\eta : X \mapsto Z$ is a 5-layer GNN encoder with $F = F_s = F_v = 32$ scalar- and vector channels and implements the translation vector, i.e. $k = 1$. The encoding network $\eta$ outputs four quantities: two SE(3) invariant node embedding matrices $\widetilde{H}, M \in \mathbb{R}^{N \times F}$, one SO(3) equivariant order-3 tensor $\widetilde{W} \in \mathbb{R}^{N \times 3 \times F}$ as well as another SE(3) equivariant matrix $T \in \mathbb{R}^{N \times 3}$.

We use two linear layers[4] to obtain the SE(3) invariant embedding matrix $H \in \mathbb{R}^{N \times 2}$ as well as the SO(3) equivariant embedding tensor $W \in \mathbb{R}^{N \times 3 \times 2}$. Notice that the linear layer returning the $W$ tensor can be regarded as the function $\psi_{\text{rot}}$ that aims to predict the group action in the SO(3) symmetry, while we use the identity map for the translation vector, i.e. $\psi_{\text{transl}} = T$.

As point clouds can be regarded as sets, we obtain an permutation invariant embedding by averaging over the first dimension of the $\{H, W, T\}$ tensors,

$$h = \frac{1}{N} \sum_{i=1}^{N} H_i \ \in \mathbb{R}^2 \ , \tag{27}$$

$$t = \frac{1}{N} \sum_{i=1}^{N} T_i \ \in \mathbb{R}^3 \ , \tag{28}$$

$$w = \frac{1}{N} \sum_{i=1}^{N} W_i \ \in \mathbb{R}^{3 \times 2} \ , \tag{29}$$

while we use the $M$ matrix to predict the permutation matrix $P_\sigma$ with the $\psi_{\text{perm}}$ function, in similar fashion as described in Eq. (16). To construct the rotation matrix $R$ out of 2 vectors in $\mathbb{R}^3$ as described in Section 3, we utilize the SO(3) equivariant embedding $w$.

The decoding network $\delta : Z \mapsto X$ is similar to the encoder a 5-layer SE(3)-equivariant GNN but does not model the translation vector, i.e. $k = 0$. The decoder $\delta$ maps the SE(3) as well as S(N)-invariant embedding $h$ back to a reconstructed point cloud $\hat{P} \in \mathbb{R}^{N \times 3}$. At the start of decoding, we utilize a linear layer to map the $G-$invariant embedding $h \in \mathbb{R}^2$ to a higher-dimension, i.e.

$$\tilde{h} = W_0 h + b_0 \ \in \mathbb{R}^{F_s}, \tag{30}$$

Next, to "break" the symmetry and provide the nodes with initial type-0 features, we utilize fixed (deterministic) positional encodings as suggested by [Winter et al. (2021)] for each node $i = 1, \ldots, N$ to be summed with $\tilde{h}$. Notice that this addition enables us to obtain distinct initial type-0 embeddings $\{\hat{h}_i\}_{i=1}^N$.

For the start positions, we implement a trainable parameter matrix $P_\theta$ of shape $(N \times 3)$ for the decoder.

Now, given an initial node embedding $\hat{H} \in \mathbb{R}^{N \times F_s}$, we apply the S(N) group action, by multiplying the predicted permutation matrix $P_\sigma$ with $\hat{H}$ from the left to obtain the canonical ordering as

$$\hat{H}_\sigma = P_\sigma \hat{H} \ . \tag{31}$$

To retrieve the correct orientation required for the pairwise-reconstruction loss, we multiply the constructed rotation matrix $R$ with the initial start position matrix $P_\theta$

$$\hat{P}_r = P_\theta R^\top . \tag{32}$$

---

[4]The transformation is always applied on the last (feature) axis.

Table 1: Comparison of our approach vs classical and quotient autoencoder (QAE) as well as an fully equivariant AE with invariant pooling after training.

| Model | Rec. Loss | KNN Acc. |
|---|---|---|
| classical | 0.0170 | 0.68 |
| QAE | 0.0227 | 0.82 |
| invariant (ours) | 0.0162 | 0.90 |
| equiv AE (norm) | 0.0189 | 0.56 |
| equiv AE (angle) | 0.0189 | 0.53 |
| equiv AE (complete) | 0.0189 | 0.67 |

With such construction, we can feed the two tensors to the decoder network $\delta$ to obtain the reconstructed point cloud as

$$\hat{P}_{\text{recon}} = \delta(\hat{H}_\sigma, \hat{P}_r) + t \,, \tag{33}$$

where $t \in \mathbb{R}^3$ is the predicted translation vector from the encoder network, added row-wise for each node position.

# E    Further Experiments

### E.1    Rotated MNIST

We also implemented and trained the quotient autoencoder (QAE) approach proposed by Mehr et al. (2018a) on the MNIST dataset for the group $\text{SO}(2)$, discretized in 36 rotations with the loss

$$\min_{\theta \in \{10i, i=0,...,35\}} \{\text{MSE}(x - \rho_X(g(\theta))y)\} \,, \tag{34}$$

where $x$ is a MNIST sample and $y$ is the reconstructed sample. We evaluated the resulting embeddings on the rotated MNIST test set (in such a way that the evaluation is the same as for our model). In Figure 7 we plot TSNE embeddings for this approach, and we can observe that the embedding space shows a clearer structure, in comparison with the classical model. However, in comparison, our approach results in a better clustering of the different digits classes. That shows that the discretization step, while it helps in structuring the embedding space in "signal clusters", still does not capture the full continuous nature of the group. To further quantitatively compare the three methods (ours, QAE and classical AE), we evaluated the reconstruction loss as well as the (digit class) classification accuracy of a KNN classifier trained on 1000 embeddings of each method. We present in the table below the results for the reconstruction loss and for the classification accuracy of a KNN classifier trained on the AE embeddings. To obtain a fair comparison, we kept the architecture and the training hyperparameters exactly identical for all the strategies. We note that our strategy outperforms both the classical AE as well as the strategy of QAE in both tasks.

In an additional experiment, we trained a fully equivariant AE (that is, the embedding itself is fully equivariant, i.e. multiple 2-dimensional vectors) on MNIST with $G = \text{SO}(2)$, followed by an invariant pooling afterwards (after the training) to extract the invariant part. Specifically, we have trained KNN classifiers on (a) the invariant embedding corresponding to the norm of the 2-dimensional vectors forming the bottleneck representation, (b) the angles between the first and all other vectors and on (c) the full invariant embedding we obtained by combining the the norms and angles. We choose the number of vectors in the bottleneck in such a way that the dimensionality of the full invariant representation coincides with the one of our model. We visualized the resulting TSNE embeddings in Figure 7 and show the downstream performance of the KNN classifiers in Table 1. From the results we can see that, in comparison to the approximate invariant (QAE) and our invariant trained model, the invariant projected equivariant representations perform inferior. Although we extract a complete invariant representation (which performs better than a subset of this representation like the norm or angle part), the resulting representation is apparently not as expressive and e.g. useful in a downstream classification task. This aligns well with our hypothesis, that our proposed framework poses a sensible supervisory signal to extract expressive invariant representations that are superior to invariant projections of equivariant features.

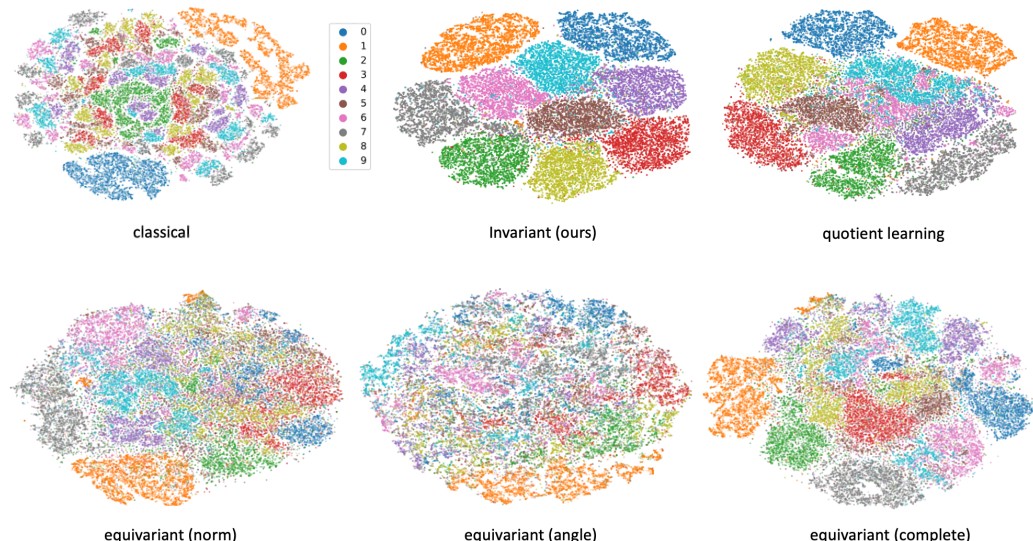

Figure 7: Top row: TSNE embedding of the encoded test dataset for a classical autoencoder, our proposed SO(2) invariant autoencoder, and for the quotient autoencoder of Mehr et al. (2018a). Bottom row: Fully equivariant trained autoencoder with invariant projection after training, either by taking the norm, angles between vectors or the combination (complete).

| Target | Fraction | Pretrained | From Scratch |
|--------|----------|------------|--------------|
| $H$ | 0.05 | 0.7529 | 0.0970 |
|  | 0.25 | 0.9908 | 0.9093 |
| $G$ | 0.05 | 0.7703 | 0.4758 |
|  | 0.25 | 0.9856 | 0.9751 |
| $U$ | 0.05 | 0.6083 | 0.2574 |
|  | 0.25 | 0.9962 | 0.9808 |
| $\langle R^2 \rangle$ | 0.05 | 0.7806 | 0.1468 |
|  | 0.25 | 0.9918 | 0.8546 |
| $\mu$ | 0.05 | 0.8698 | 0.8443 |
|  | 0.25 | 0.9718 | 0.9718 |
| $\alpha$ | 0.05 | 0.9455 | 0.9237 |
|  | 0.25 | 0.9937 | 0.9764 |

Table 2: Generalization performance in terms of the coefficient of determination $R^2$ of models on a held-out test set of 1000 samples. Higher $R^2$ indicates better performance.

## E.2 QM9

For the QM9 dataset, we use the same model components as described in the Tetris experiment, with the difference of including atom species as $SE(3)-$invariant features and setting $F_s = 256, F_v = 32$ and increasing the dimensionality of the latent space to 256.

### E.2.1 Finetuning

We performed additional experiments on the pretrained group-invariant AE on the extended GEOM-QM9 dataset Axelrod & Gomez-Bombarelli (2021) which, as opposed to the standard QM9 dataset ($\approx 130k$ samples), contains multiple conformations of small molecules. We trained the autoencoder on a reduced set of GEOM-QM9 ($\approx 641k$), containing up to 10 conformations per molecule and utilized this pretrained encoder network to regress (invariant) energy targets, such as internal energy $U$ or enthalpy $H$ on the original QM9 dataset.

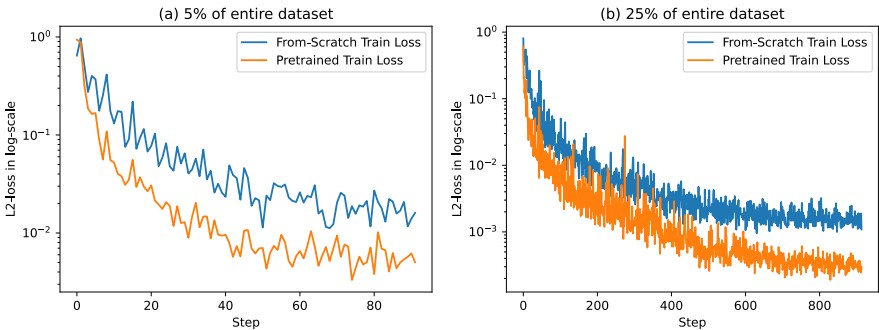

Figure 8: Learning curves of models trained on limited enthalpy $H$ targets.

We observed that the pretrained encoder network learns faster and achieves better generalization performance than the architectural identical network trained from scratch. In Figure 8 we illustrate the learning curves for the two networks on different fraction on $5\%$ and $25\%$ labelled samples from the original QM9 dataset to analzye the benefit of finetuning a pre-trained encoder network in a low-data regime, when regressing on the enthalpy $H$. On a held-out test dataset of $1000$ samples, the pretrained encoder network achieves superior generalization performance in terms of $R^2$ with $0.7529$ vs. $0.0970$ in the $5\%$ data regime, and $0.9908$ vs. $0.9093$ in the $25\%$ data regime compared to the encoder that was trained from scratch. In Table 2 we show additional comparisons of the pretrained network against a network that was trained from scratch for 50 epochs on the restricted dataset.

As shown in Table 2, the pretrained encoder achieves improved generalization performance on the test dataset compared to its architectural identical model that was trained from scratch. We believe that training the group-invariant autoencoder on a larger diverse dataset of (high-quality) molecular conformations facilitates new opportunities in robust finetuning on different data-scarse datasets for molecular property prediction.

### E.2.2   Molecular Conformations: Further Examples

We show additional reconstructions of 12 randomly selected small molecules from the QM9 test dataset. Noticeably, our trained autoencoder is able to reconstruct molecular conformations with complex geometries as depicted in the third column (from the left). We notice that the AE is not able to perfectly reconstruct the conformation shown in the 4th column of the 2nd row. Although this molecule does not exhibit a complicated geometrical structure, its atomistic composition (of only containing nitrogen and carbon as heavy atoms) could be the reason why the encoding of the conformation is pointing into a non-densely populated region in the latent space, as nitrogen does not have a large count in the total QM9 database, see Figure 10.

Training was done on one NVIDIA Tesla V100 GPU in approximately 1 day.

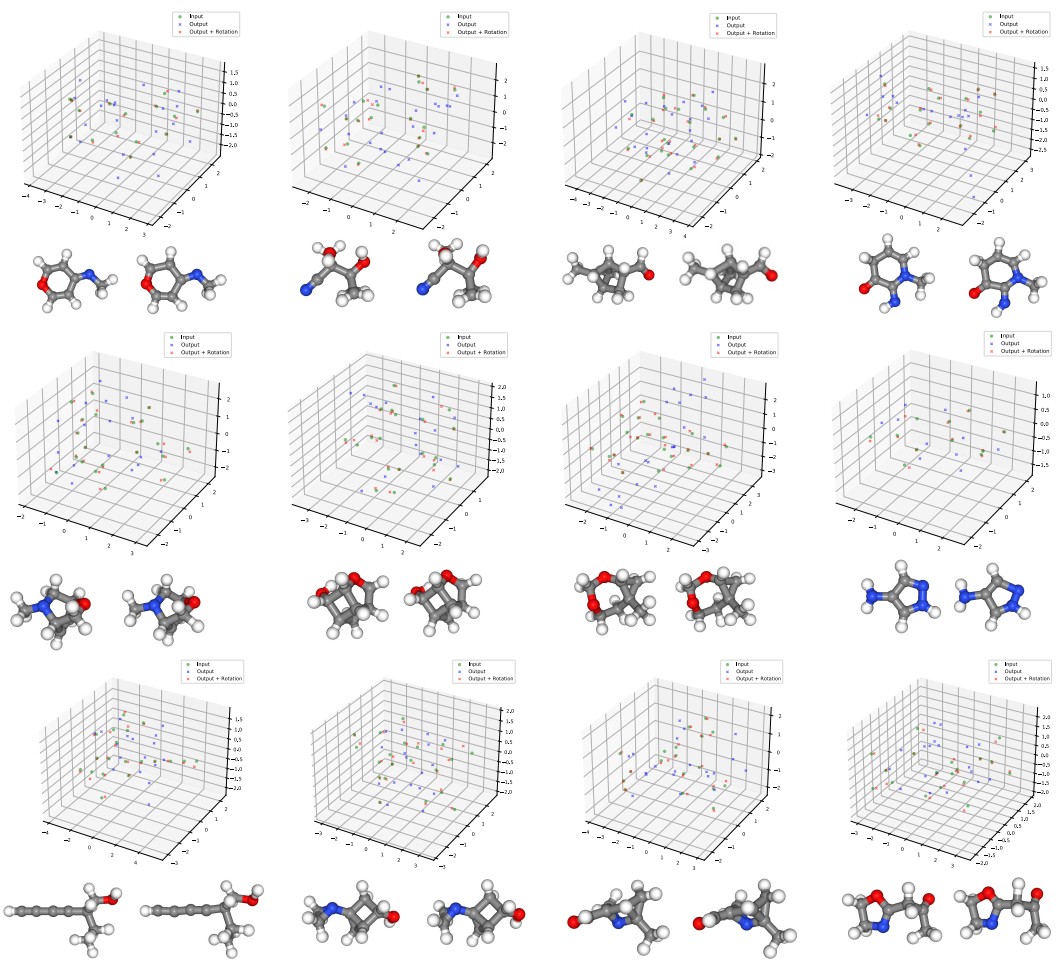

Figure 9: 12 molecular conformations and their reconstructions represented as point cloud and ball-and-stick model (left true, right predicted).

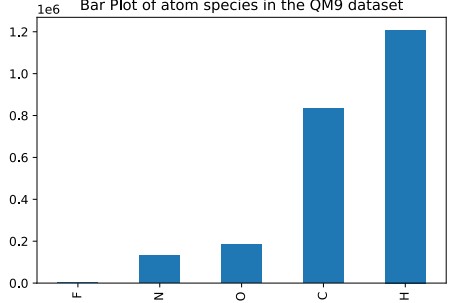

Figure 10: Atomistic species count on the QM9 dataset.

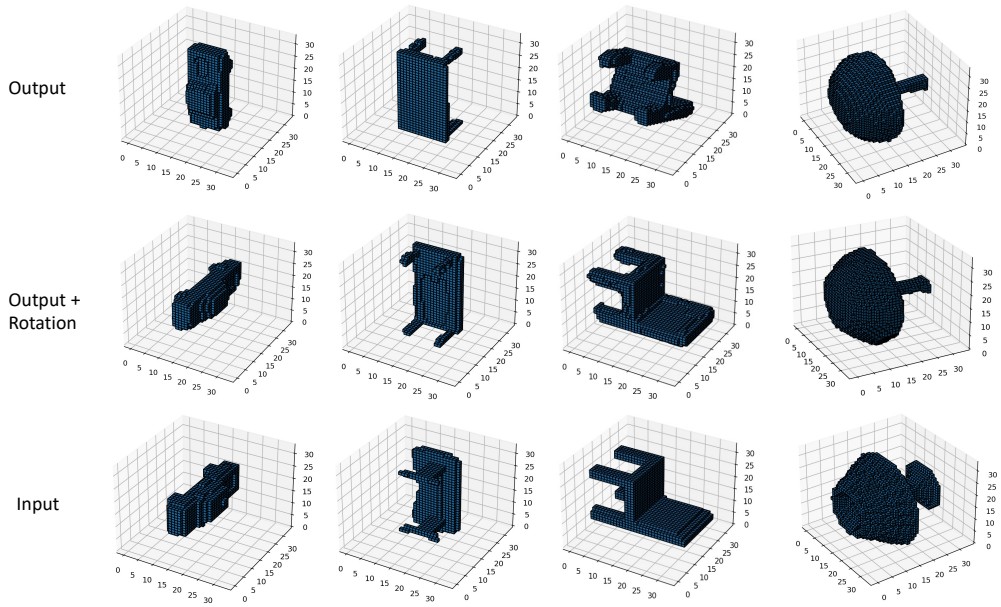

Figure 11: Example reconstructions from our proposed method trained on the voxelized ShapeNet dataset Chang et al. (2015).