# OpenReview forum: "Unsupervised Learning of Group Invariant and Equivariant Representations"
_NeurIPS.cc/2022/Conference — NeurIPS 2022 Accept_

### Official Review · Reviewer_4UN9 · 2022-06-30

**Rating:** 6
**Confidence:** 4
**Soundness:** 3 good
**Presentation:** 3 good
**Contribution:** 2 fair

**Summary:**

Equivariant neural networks have been shown to improve data efficiency and model performance, but require knowledge about the group acting on the data space to be implemented. This work promises to learn the group action from data instead, foregoing the need for ad hoc implementations of the group action on the feature maps / data space. To this end, authors propose learning two separate mappings: (1) from the input sample to a group-invariant canonical representation and (2) from the input sample to a group element / group action. Together, these allow for reconstructing the input sample under its observed pose. The authors show, in a range of experiments, that their framework is able to learn to successfully reconstruct input samples under their respective poses, from an invariant embedding and a learned group action.

**Questions:**

1. I would like to see the authors address my concern regarding their claim on avoiding the necessity of ad-hoc group-specific implementations. Is it not correct that you need such group-specific implementations of $\eta, \mu$ and $\xi$ in your framework? If so, how does your framework reflect the claim that you are "learning the group action"?

2. I don't fully understand the need for learning separate encoders $\eta$ and $\mu$. Can you not reuse the same embedding in both cases and to obtain $z$ simply group invariantly project over the equivariant embedding?

3. I'm not completely clear on the need for separating equivariant and invariant representations in the first place, could the authors expand on this a bit? Is it simply because we want to place no constraints on the decoder $\delta$ that we discard information on group action over $X$ in the $\eta\rightarrow\delta$ pathway? But if we want a latent space structured according to a group action, can we not simply use an equivariant encoder and decoder?

4. Regarding the experiments, in 5.1 you mention that, upon inspection, the encoded latent representation is indeed rotation invariant up to machine error for 90 deg rotations, and shows interpolation artefacts otherwise. But this is just a consequence of the method of interpolation you are using to rotate the images and the encoder (SO(2)-CNN) you are using, and isn't necessarily a result of your framework correct? You use an encoder which is invariant, hence the learned representation is invariant. I also believe the invariance property of your encoder completely explains the rotation invariant structure of your latent space. To me this doesn't seem a result of your framework, but of the (group-specific) encoder you are using?

5. For the set of digits experiment in 5.2, the experimental setup is not clear to me. Is the task permutation invariant or equivariant? Do you only evaluate the presence of digits in the reconstruction or their order as well? If you are only evaluating the set contents, is the equivariant pipeline discarded for this experiment? Or, if you are evaluating their order as well, I think Fig 4a is misleading, as in addition to the embedding your framework needs the permutation matrix for reconstruction (which is of larger dimensionality than the embedding).

6. In experiment 5.3 you investigate a SE(3) and S_N autoencoder. From line 322, it seems you are only evaluating your reconstruction up to to a translation and permutation (you are only predicting a rotation with $\psi$)? So the autoencoder as a whole is invariant to permutations and translations and only equivariant to rotations, correct? In which case, why didn't you make it invariant to translations and permutations as well? As your autoencoder is invariant to translation, why do you apply translation augmentations to the input?

7. In line 330 you note that your SE(3)-equivariant model (which it is not, it is only SO(3)-equivariant) can distinguish between mirrored chiral shapes, and that SE(3)-invariant models are unable to do so. I believe this is incorrect; reflections are part of the E(3) group, and not of the SE(3) group, hence SE(3)-invariant models should be able to distinguish between these shapes. Could you comment on this?


**General post-rebuttal response**
I would like to thank the authors for their very thorough rebuttal, addressing most of my concerns, and the concerns raised by the other reviewers. I think the work significantly improved in experimental evaluation and comparison. As indicated in my last comments, parts of the motivation and framing of the model proposed by the authors remains misleading / incorrect to me. I would recommend that, were this work to be accepted, the authors review some of the claims they make in the paper regarding the "learning of a group action". I feel such claims should be addressed before publication.

The authors did address many of my questions regarding the broad motivation for their invariant-equivariant approach, but from my point of view, the authors skip a step by discarding fully invariant aggregation of equivariant self-supervised representations. I would recommend the authors empirically or theoretically show that such representations don't work in practice, as motivation for their (arguably more complex) approach.

On the condition of inclusion of their responses in an updated version of the manuscript, I raise my recommendation to a borderline accept.



**Limitations:**

I did not see the authors address any limitations of their work.

**Strengths And Weaknesses:**

**Originality and Significance** The authors propose a sound method for implementing group equivariant auto-encoders. I have some concerns regarding claims the authors make throughout regarding learning the group action. If my understanding is correct, this method only "learns" the group action on the data space in the sense that the $\mu$ operator learns a map from any point in the data space $x \in X$ into an embedding space $Y$, where this map is required to be equivariant and $Y$ is required to be a homogeneous space of the group. Next, an operator $\xi$ essentially maps this homogeneous space into the left-regular group action. The whole point of this encoder is to extract pose, but since you presuppose the encoder to be equivariant to the group action, is this not trivial?

Since $\mu$ is equivariant to the group action it inherently preserves pose, i.e. prop 2.7 is a property of any operator equivariant to the group action, and therefore the group action isn't actually "learned"; its structure in the embedding space is given by the equivariant operator. Furthermore, this separation between equivariant and invariant representations hardly seems like a contribution; as you yourself mention, most applications of equivariant neural networks learn equivariant representations which are then projected to an invariant representation. I don't fully understand the need for learning separate encoders $\eta$ and $\mu$. Can you not reuse the same embedding in both cases and simply apply a symmetric operation to $\mu(x)$ obtain to obtain $z$? In fact, for experiment 5.1 you mention that you use the same architecture for both pipelines, why not share the representation?

I would like to see the authors note on how this work relates to Keller & Welling (2021). The methods seem related in that both authors propose learning group equivariant latent representations from data without explicitly encoding the group action.

**Quality and Clarity** The authors explain their derivations in thorough and clear manner. In section 3 I'm left with some questions. [line 186, 187] "*we turn to describing our construction of $\xi, \mu$ and $Y$ for a number of groups*". For ${\rm SO(2)}$ you only describe how to construct $Y$ and $\xi$, what about $\mu$? For ${\rm SO(3)}$ you only describe $Y$, what about $\xi, \mu$? Same for the rest of the examples. Could you address this?

$\xi$ needs to be constructed by hand, i.e. a new group requires a new implementation of $\xi$ AND a new implementation of $\eta$ and $\mu$ because of new equivariance constraints. There seems to be no actual learning of the group action and although you claim your method "avoids the necessity of ad-hoc group-specific implementations" it does in fact require such group-specific implementations to be implemented itself?

From my view, the experiments used by the authors to investigate and evaluate their approach do not adequately show its merits. For details on my concerns, see below questions 4-7. The experiments do not convince me of the need for separate invariant-equivariant terms. To this end, could the authors, in their experimental setups, compare with a fully equivariant encoder-decoder?

*References*

Keller, T. A., & Welling, M. (2021). Topographic vaes learn equivariant capsules. Advances in Neural Information Processing Systems, 34, 28585-28597.

---

> ### Author Response · Authors · 2022-08-02
> **Response 1: General Motivation**
>
> We thank the reviewer for their valuable comments. Below you will find our responses to your comments.
>
> We take the opportunity here and perhaps clarify some doubts the reviewer raised regarding our motivation. Our main goal is to design an effective method to **learn the most powerful invariant embeddings in an unsupervised manner utilizing equivariant neural networks**. To this end, the decoder $\delta$ and group function $\psi$ are mainly used as auxiliary tool to enable the training of such an invariant encoder.
>
> 1. **Why invariant?** Many interesting properties are invariant to certain group actions (e.g. ground state energy of a molecular conformation is invariant to rotations/translations of the molecule). The advantage of invariant representations for e.g. a downstream model is the necessity of less data, no need for augmentation, and the fact that the model can focus on the actual signal/shape and not its pose.
> 2. **Why equivariant?** We still want to extract as most information as possible out of the data, and therefore we do not wish to restrict ourselves to (a) invariant input features or (b) models that are just a compositions of invariant functions.
> 3. **Why Autoencoders?** We do not want to rely on labeled data as supervisory signal to extract expressive invariant representations but use unsupervised learning. Autoencoders (AE) are a straight-forward approach to extract expressive high-level features from data. However, as AEs are trained on a reconstruction loss in the non-invariant data domain, we propose our framework to still enable encoding of invariant features by separating the invariant (shape) from the equivariant (pose) part.
>
> Hence, as the reviewer suggest to "compare with a fully equivariant encoder-decoder" would be out of scope of this work, since a trained equivariant encoder-decoder cannot afterwards be used to extract expressive invariant embeddings (as discussed below).

---

> ### Author Response · Authors · 2022-08-02
> **Response 2: Why separate invariant and equivariant parts?**
>
> The reviewer raises multiple times their concern about the separation of the invariant and equivariant encoders and representations. We are not completely sure if the reviewer is concerned about either (a) the necessity of invariant part during training, i.e., whether we can train the autoencoder in a purely equivariant way and apply an invariant projection on the equivariant representation after the training to extract invariant embeddings or (b) the utilization of two different networks and weights for the equivariant ($\psi$) and invariant ($\eta$) part of of our framework.
>
> For (a) we would like to emphasize that training purely in an equivariant way and symmetrizing after training is in general not straight-forward to do and will not lead to expressive representations. Let's discuss the example of the rotation group (e.g. $\text{SO}(2)$). If we train an autoencoder with an equivariant bottleneck (e.g. vectors) we can make the representation after the training rotation invariant by e.g. only consider the norm of the embedding vectors. However, if the autoencoder is able to perfectly reconstruct the inputs from the equivariant embedding vectors, it is easy to see that the norm of the vectors will not necessarily store the information needed to reconstruct the input up to a rotation nor do we have a decoding function that is trained to do so. By separating the embedding into a invariant and equivariant part during training, we force the model to store all invariant information (shape) in the invariant embedding and train a decoder that can reconstruct the input (up to a rotation) from these invariant features. The equivariant part is only used to align input and decoded elements to enable optimization by a reconstruction objective.
>
> For (b), we are in fact sharing the same network for the invariant and equivariant part and extract the invariant information by applying a invariant projection (e.g. taking the norm of rotation equivariant vectors) as proposed by the reviewer (also compare answer to question 2). However, crucially we are doing this during the training. Hence the network has to learn to store meaningful information in the invariant projections (e.g. norm of a vector) to minimize the reconstruction loss.
>
> The reviewer also notes that
>
> >...this separation between equivariant and invariant representations hardly seems like a contribution; as you yourself mention, most applications of equivariant neural networks learn equivariant representations which are then projected to an invariant representation.
>
> Please note that in order to extract meaningful invariant representations by projecting invariantly from an equivariant representation one needs some kind of supervision. Most works do this by utilizing labeled data in a supervised learning paradigm (i.e. input -> equivariant model -> invariant projection -> invariant label). In unsupervised learning we dont have access to such labels. Here we utilize the reconstruction loss of an autoencoder as supervision signal, however, as discussed and motivated in our manuscript this loss is non-invariant (unlike in supervised learning case), which we tackle by our proposed method. Hence we believe our work poses a major contribution, as it allows to train invariant feature extractor in an unsupervised way.
>
> The reviewer also writes:
>
> >The whole point of this encoder $[\psi]$ is to extract pose, but since you presuppose the encoder to be equivariant to the group action, is this not trivial?
>
> This is not quite correct. As addressed below in the questions, the point of $\psi$ is not learning a absolute pose (which in some sense would be indeed trivial because of equivariance) but the pose with respect to the (unknown to the encoder) canonical (invariant) pose of the reconstructed sample (which is not trivial and subject to the joined learning process of $\eta$, $\delta$ and $\psi$). The equivariance property merely ensures that this needs to be learned once per orbit (and not separately for each elements of the orbit).

---

> > ### Comment · Reviewer_4UN9 · 2022-08-08
> > **The need for equivariant and invariant parts**
> >
> > Thank you for clarifying, I believe I now understand your motivation for separate equivariant and invariant pipelines in the self-supervised setting. You would like to learn transformation-invariant representations, but doing this in a self-supervised setting with a reconstruction task requires an autoencoder equivariant to this transformation. To have supervision for the invariant representation, you argue for an invariant reconstruction followed by a transformation to obtain an equivariant reconstruction. Indeed, this seems sensible.
> >
> > > However, if the autoencoder is able to perfectly reconstruct the inputs from the equivariant embedding vectors, it is easy to see that the norm of the vectors will not necessarily store the information needed to reconstruct the input up to a rotation nor do we have a decoding function that is trained to do so.
> >
> > It took me a bit to understand your argument against simply using an equivariant autoencoder and invariantly aggregating the equivariant latent representation, but I believe you argue that this does not lead to expressive invariant representations. This does not necessarily seem trivial to me, and I would like to see some more extensive (either empirical or theoretical) motivation here. I believe in the above example the norm of the vector would in fact store all information needed to reconstruct the input invariantly; information on the pose of the input is only stored implicitly, in the direction of the embedding vector.

---

> > > ### Author Response · Authors · 2022-08-08
> > > **Response 1: Theoretical Motivation**
> > >
> > > We thank the reviewer for this response. We want to provide additional evidence that the two following approaches are very different regarding the expressiveness of the invariant representation $z$:
> > >
> > > - Equivariant AE with pooling **after** training: In this approach the full AE is equivariant, and can be trained with no necessity of extra strategies since the loss is automatically invariant. The bottleneck representation $z'$ is equivariant with respect to $G$, and in order to extract an invariant representation $z$ we can apply an invariant pooling $z = p(z')$.
> > >
> > > - Equivariant AE with pooling **during** training: In this approach the AE is equivariant, but an invariant pooling $z = p(z')$ is applied in the network during training. This means, all the information flow (up to a group transformation) passes through the invariant representation $z$ during training.
> > >
> > > If we understand correctly, the reviewer is concerned that Approach 1 might result in similarly expressive representations as (more sophisticated) Approach 2, which we follow in our work. That is, we want to leverage the expressiveness of equivariant models (encoder and decoder functions), but at the same time force all the information to flow through an invariant bottleneck, to force the model to store all the relevant information necessary to reconstruct the input in an invariant fashion.
> > >
> > > We can show mathematically that, given one function $p$, it is often not enough to capture the full information stored in an equivariant representation.
> > > For example, given two vectors $\mathbf{r}_1 = (x_1, y_1), \mathbf{r}_2=(x_2, y_2)$ in 2D and for group $G = \text{SO}(2)$, taking as our invariant pooling the norm of the two vectors
> > >
> > > $z = (||\mathbf{r}_1||, ||\mathbf{r}_2||) = (\sqrt{x_1^2+y_1^2}, \sqrt{x_2^2 +y_2^2})$~,
> > >
> > > does not allow the full system reconstruction. To do that, we would need to know the angles between the two vectors as well.
> > >
> > > Theoretically, it would be possible to extract a complete invariant representation from an equivariant one. This would imply extracting all possible independent pooling functions. In the above example, these corresponds to $||\mathbf{r}_1||$, $||\mathbf{r}_2||$, $\mathbf{r}_1 \cdot \mathbf{r}_2$. However, for higher dimensional and more complex data and general groups it is unfeasible to extract all possible invariant pooling. On the other hand, when we choose a specific subset of pooling, we will inevitably loose some amount of information stored in the equivariant representations.
> > >
> > > Our methods avoids this problem since the model learns to solve the reconstruction task with the specific subset of pooling we implement in the architecture. Thus, even if our invariant embedding will loose dimensionality with respect to the equivariant representation, we are guaranteed that it is enough for the reconstruction if the model is trained successfully.
> > >
> > > We can extend the above to any number of vectors in 2D with $\text{SO}(2)$ symmetry. Let us suppose we have $M$ vectors
> > > $\mathbf{r}_i=(x_i, y_i)$ for $i=1,\dots,M$. We can completely determine the system up to a global rotation by considering, as before, the $M$ norms $||\mathbf{r}_i||$ as well as all the angles (determined by the scalar products) between the vectors $\mathbf{r}\_{2, \dots,M}$ and $\mathbf{r}_1$, which we represent as $\mathbf{r}\_{2,\dots,M}\cdot \mathbf{r}_1$. These correspond in total to $2M-1$ parameters, namely, $M$ norms and $M-1$ angles. This is the complete invariant representation, as the only parameter missing to recover the full $2M$ Cartesian system is one parameter corresponding to the angle for a global
> > > $\text{SO}(2)$ rotation.
> > >
> > > As we have seen, while this is indeed possible for simple and smaller groups, it is still tedious and rather quickly impractical for larger groups (or product of groups).

---

> > > > ### Comment · Reviewer_4UN9 · 2022-08-09
> > > > **Theoretical Motivation**
> > > >
> > > > Thanks for addressing my concerns regarding the motivation for your approach so thoroughly. Although my concerns regarding a number of statements in the manuscript remain, the authors have addressed most of my concerns. I've reflected this in my recommendation.

---

> > > ### Author Response · Authors · 2022-08-08
> > > **Response 2: Empirical Motivation**
> > >
> > > Following the very meaningful suggestion of the reviewer, we performed an additional experiment, where we compare
> > > the approaches listed above in the theoretical motivation. Namely, we trained a fully equivariant AE (that is, the embedding itself is fully equivariant, i.e. multiple 2-dimensional vectors) on MNIST with $G=\text{SO}(2)$, and we perform an invariant pooling afterwards to extract the invariant part.
> > > Specifically, following the discussion above, we have trained KNN classifiers on (a) the invariant embedding corresponding to the norm of the 2-dimensional vectors forming the bottleneck representation, (b) the angles  between the first and all other vectors and on (c) the full invariant embedding we obtained by combining the the norms and angles. We choose the number of vectors in the bottleneck in such a way that the dimensionality of the full invariant representation coincides with the one of our model. We visualized the resulting TSNE embeddings in Appendix B (Figure 6) of the newly revised manuscript. We also updated the corresponding table in the revised manuscript (Appendix B Table 1) as follows
> > >
> > > | Model      | Rec. Loss | KNN Acc.
> > > | ----------- | ----------- | ----------- |
> > > | classical      | 0.0170       | 0.68
> > > | QAE   | 0.0227        | 0.82
> > > | ours   | 0.0162       | 0.90
> > > | equiv AE (norm)      | 0.0189       | 0.56
> > > | equiv AE (angle   | 0.0189        | 0.53
> > > | equiv AE (complete)   | 0.0189       | 0.67
> > >
> > > From the results we can see that, in comparison to the approximate invariant (QAE) and our invariant trained model, the invariant projected equivariant representations perform inferior. Although, as discussed above, we extract a complete invariant representation (which performs better than a subset of this representation like the norm or angle part), the resulting representation is apparently not as expressive and e.g. useful in a downstream classification task. This aligns well with our hypothesis, that our proposed framework poses a sensible supervisory signal to extract expressive and higher-level (e.g. well-structured according to the signal such as the digit class) invariant representations that are superior to invariant projections of equivariant features. We belief that this experiment gives strong empirical evidence for the motivation of this work and thank the reviewer very much for suggesting such a comparison, strengthening our manuscript.

---

> > > > ### Comment · Reviewer_4UN9 · 2022-08-09
> > > > **The need for equivariant and invariant parts**
> > > >
> > > > Thank you for looking into this. I, too, think that this experiment adds to the motivation for your approach and strengthens your submission.

---

> ### Author Response · Authors · 2022-08-02
> **Response 3: Questions 1-5**
>
> 1. We thank the reviewer for giving up the opportunity to clarify what we mean with "learning the group action". The map $\psi: X\rightarrow G$ is a map from the data space $X$ to the group $G$. At the beginning of the training, that is, for a randomly initialized network, $\psi$ will randomly map data samples to group elements. Given our requirement that $\psi$ needs to be equivariant, the group relationship will be maintained by $\psi$, that is, if $x'=\rho_X(g')(x)$, then $\psi(x') = g\cdot \psi(x)$. Now, during training, $\psi$ needs to learn to map a sample point $x$ to the group element $g_x=\psi(x)\in G$ such that $\rho_X(g_x)\delta(\eta(x)) = x$, that is, the group element that maps the (invariant) decoded output to the (non-invariant) input.  Now, in order to achieve that we are faced with the questions: What are the conditions that such a map $\psi$ needs to satisfy and more specifically, how do we implement it in a "systematic" fashion?
>  In Section 2.3 we derived such sufficient conditions. Further we propose an explicit construction which makes use of the two maps $\mu$ and $\xi$ and of the intermediate space $Y$. This construction satisfy all the requirement and it is group-agnostic, but it is for sure not the only one. This is where our claim that our construction works for any group comes from, which perhaps might leads to a misunderstanding. Specifically, what we mean is as follows: The mathematical property of the map $\mu$ and $\xi$, defined in Property 2.6, are general and valid for any group $G$.
>     We agree with the reviwer that the specific form, of the space $Y$ will need to depend on the group $G$, since homogeneity is a property that is group-dependent. Moreover, note that our framework is in general flexible enough to allow the use of already existing group-equivariant neural network implementations for the encoder ($\eta$ and $\mu$).
>
> 2. The reviewer is completely correct. In fact, this is what we often do in practice, that is, we have one single equivariant encoder, and the invariant bottleneck is obtained through a pooling of the equivariant output. We merely depicted the architecture in such a way in the methods section because in our opinion it makes the general learning framework clearer. However, note that we still have to separate the invariant from the equivariant part after encoding, as discussed above, to train a powerful invariant feature extractor. Also please note, that we only pass on as few equivariant features as needed to construct the group action (e.g. only one 2-dimensional vector for group SO(2)), while the invariant part might be higher dimensional (e.g. 32 dimensions in our MNIST experiment). No additional information other than a group action is passed on by $\psi$.
>
> 3. As discussed above, our main motivation in this work is to train an unsupervised model on extracting expressive representations of data that are invariant under transformation of a defined group (We do not "want a latent space structured according to a group action" as mentioned by the reviewer). In essence, the decoder $\delta$ and group function $\psi$ are just a means to this end. (a) If we are just interest in reconstruction/generation we could use a completely equivariant model (with an equivariant embedding). (b) If we would have access to labeled data, we could use a supervised model with an equivariant encoder followed by a group invariant projection. However, in our unsupervised setting, the learning signal is a non-invariant reconstruction objective. In order to extract an invariant representation, we have to separate the input representation into a invariant part (our desired output) and a equivariant part that is merely used to transform the decoded canonical element in order to evaluate the non-invariant reconstruction objective.
>
> 4. Yes, the reviewer is correct in that the encoder is invariant by design (as a consequence of using a group-specific invariant model) and the deviations discussed in 5.2 are results of interpolation artifacts. However, we want to emphasize that the expressiveness of the resulting invariant representations are consequence of our framework and not a trivial outcome of an invariant encoder.
>
> 5. The task described in Section 5.2 (set of digits) is permutation equivariant, that is, the order of the digits is relevant in the reconstruction. It is true that for reconstruction we need in addition to the invariant embedding (which effectively stores the composition as visualized in Figure 4c) the permutation matrix (visualized in Figure 4b). As our goal is the invariant representation of a set and the equivariant part (permutation matrix) is only used during training we do not count this part in Figure 4a. However, we acknowledge that Figure 4a might indeed be misleading and we will add a comment regarding this in the manuscript. Thank you for pointing this out.

---

> > ### Comment · Reviewer_4UN9 · 2022-08-08
> > **Response to answers**
> >
> > Thanks for your patient answers to my questions. Regarding questions 2, 3, and 5, my concerns have been addressed.
> >
> > Your answer to question 1: I still believe the current version of your manuscript contains misleading / incorrect statements which need to be addressed prior to publication. In multiple passages you make it seem like your model learns the group action itself. Instead, your model learns a pose estimation, right? What's more, to learn this pose estimation you need the ad-hoc group-specific implementations you claim to do away with. E.g. your abstract contains the following line:
> >
> > > The key idea is that the network learns the group action on the data space and thus is able to solve the reconstruction task from an invariant data representation hence avoiding the necessity of ad-hoc group-specific implementations.
> >
> > I don't see how you can square this line with section 3, in which you list multiple such group-specific implementations. Before publication, I would recommend the authors weaken these statements about the learning of the group action, and be transparent about the limits/requirements of their pipeline.
> >
> > Your answer to question 4: Thanks for clarifying, I would recommend you make this clear in your manuscript as well, to avoid confusion with future readers.

---

> > > ### Author Response · Authors · 2022-08-08
> > > **Response**
> > >
> > > > Before publication, I would recommend the authors weaken these statements about the learning of the group action, and be transparent about the limits/requirements of their pipeline.
> > >
> > > Agreed. We will replace the mentioned sentence from the abstract with the following sentence:
> > >
> > > *The key idea is that the network learns to encode and decode data to and from a group-invariant representation by additionally learning to predict the appropriate group action to align input and output pose to solve the reconstruction task.*
> > >
> > > Moreover, we will be more clear about what we mean with "learning the group action" in the final manuscript, i.e. adding another sentence clarify that we don't learn the group itself but the transformation of an (a priori) known group to align the input and (learned) canonical pose.
> > >
> > > > Your answer to question 4: Thanks for clarifying, I would recommend you make this clear in your manuscript as well, to avoid confusion with future readers.
> > >
> > > Agreed. We will clarify this in the revision.
> > >
> > > We thank the the reviewer for raising these concerns helping us to avoid confusion.

---

> > > > ### Comment · Reviewer_4UN9 · 2022-08-09
> > > > **Response**
> > > >
> > > > Thank you! I've updated my recommendation.

---

> ### Author Response · Authors · 2022-08-02
> **Response 4: Questions 6-7 and missing description of Xi, Mu and Y**
>
> 6. In the two Experiments of 5.3, we train an $S_N$ and SE$(3)$ equivariant autoencoder but the bottleneck $z$ is $G-$invariant as it should capture the intrinsic information of the data. For the reconstruction, we in fact, restore the set-elements with the correct orientation (i.e. considering translation and rotation) as well as correct indexing (i.e. with the right order). Hence, we predict translation vector $t\in\mathbb{R}^3$, rotation matrix $R\in$ SO$(3)$ as well as the permutation matrix $P_{\pi} = (p_{ij}) \in \mathbb{R}^{N \times N}$ which are applied on the decoder outputs, to perform the `alignment' and subsequently the reconstruction. In Figure 5, we only illustrate reconstructions after applying the rotation to emphasize the visual effect of the alignment. We thank the reviewer for raising this question and we will clarify this point in the updated manuscript.
> 7. The reviewer is correct is stating that a general SE-$(3)$-invariant model is able to distinguish such chiral shapes. We just wanted to point out that a class of such invariant models, namely the ones that restrict themselves to only considering distances between points, cannot distinguish the two shapes. In fact, the relative distance matrix for the two shape is identical, as there is no difference whether the last point is on the left or on the right. Obviously, an invariant model which also takes angles into consideration would be able to distinguish such shapes. The reason why we made such a comment in the text is to point out that the correct application of our approach avoids the risk to encounter such problems, since the autoencoder must be able to reconstruct the whole data space, while invariant ad-hoc models, even if very successful like in the case of SchNet, which were designed for ad-hoc task, suffer from lack of generalization when applied to different datasets.
>
> ### Description of $Y, \mu, \xi$  for $\text{SO}(n)$ and $T_n$
> We thank the reviewer for pointing out that that we are not perhaps always explicit enough in describing the various maps determining the function $\psi$. We will complement this here, and we will make sure to add these additional details in a separate appendix section.
>
> #### $\text{SO}(n)$
>
> We describe here the general procedure for $\text{SO}(n)$, which automatically includes the cases $\text{SO}(2)$ and $\text{SO}(3)$.
>
> The space $Y$ is $S^{n+1}$, the $n+1$-dimensional sphere. We obtain a point y in $Y$ through the map $\mu$, which predicts $n$ $n$-dimensional vectors and take the various ortho-normal combinations, so to obtain $n$ orthonormal vectors $\widehat{y}_i$ in $S^{n+1}$. The collection of such vectors . By choosing $y_0=I_n$, the $n\times n$ unit matrix, the map $\xi$ assigns to a point
>
> $ y = \begin{pmatrix}
>         \widehat{y}_1 & \widehat{y}_2 & \cdots & \widehat{y}_n
>     \end{pmatrix} $
>
> the group element $g\in  \text{SO}(n)$, such that
>
> $
>     \rho_Y(g)y_0 = y~.
> $
>
> This indeed, by construction, corresponds to rotation element whose rotation matrix is given exactly by $y$ defined in the equation above.
>
> #### $T_n$
>
> Here $Y=\mathbb{R}^n$, and $\mu$ predicts simply a $n$-dimensional vector in $\mathbb{R}^n$. In all our experiments, we take $y_0=\mathbf{0}$, the origin of $\mathbb{R}^n$. Thus, $\xi$ is the map that assign to each element $y\in Y$ the group element $g\in T_n$ such that $y=y+ \mathbf{0} = y + y_0$. In this sense, since $T_n=\mathbb{R}^n$, $\xi$ is the identity function, $\xi(y)=y$.

---

> > ### Comment · Reviewer_4UN9 · 2022-08-08
> > **Response to answers**
> >
> > Thanks for your patient answers to my questions. Regarding my 7th question, I see I must have misunderstood this line in your manuscript; I interpreted your comment in line 335 to mean that any SE(3) invariant models would be unable to distinguish these chiral shapes, but you only state that these specific instances of SE(3) invariant models are unable to do so. I would recommend making this point clear; SE(3) invariance does not necessarily prevent distinguishing chiral shapes.

---

> > > ### Author Response · Authors · 2022-08-08
> > > **Response**
> > >
> > > We totally agree. We will clarify this point in the final manuscript. We thank the reviewer for making us aware of this.

---

> ### Author Response · Authors · 2022-08-08
> **General Response to the Revised Review**
>
> We thank the reviewer again for responding to our rebuttal and clarifying some of the concerns not properly addressed by us. Below we posted detailed answers to different comments made by the reviewer.
>
> Most notably, we added another theoretical and empirical justification for our motivation to separate the invariant and equivariant part in our framework in order to learn to extract expressive invariant representations of data. We think that these additional responses and experiments further clarify and underline the contribution and impact of our work and hope that the reviewer reconsiders their evaluation of our work.

---

### Official Review · Reviewer_FQV4 · 2022-07-02

**Rating:** 6
**Confidence:** 4
**Soundness:** 3 good
**Presentation:** 3 good
**Contribution:** 3 good

**Summary:**

An autoencoder architecture that is based on equivariant and invariant neural layers is proposed. The most important aspect of the architecture is that the encoder stage can map the input to a guaranteed group invariant feature and a group action such that the decoder operates on the group invariant feature and produces the reconstruction of the input in a canonical form. The predicted group action is then applied on this canonical decoder output to reconstruct the input exactly (assuming the autoencoder is trained well). Experiments on images and point clouds are performed for a variety of groups.

**Questions:**

The authors should discuss what, if any, are the downstream applications of such a method. Is it data compression? Or do they envision the group invariant features learned by their method to be useful for high-level tasks like robust classification?

Related to reference (a) above, if the eventual goal is to use these group-invariant features for tasks like classification, I wonder why we need to find the group action at all. If we have a group-invariant latent feature, we can use a decoder and a group invariant reconstruction loss for training the autoencoder.

I do hope that the authors address the weaknesses in the paper. Particularly, I would like them to clarify the contributions of the paper better, especially in light of other closely related work already in literature. Experiments need to be improved as well, as I have indicated.

**Limitations:**

The authors have not discussed the limitations of this work.

**Strengths And Weaknesses:**

Strengths:

1. The method provides guaranteed invariance to the transformation group.
2. The method is general enough to apply to both discrete (such as permutations) and continuous (such as rotations and translations) groups, and to a diverse set of domains -- images, point clouds.
3. Experiments show that the network is indeed able to learn a group invariant and the corresponding group action when trained with the reconstruction loss alone.


Weaknesses:

1. Missing related work: I think the main weakness in the paper is that the authors have missed plenty of related work in literature. I give some examples here of works which consider unsupervised learning of group invariant representations:

(a) Manifold Learning in Quotient Spaces: https://openaccess.thecvf.com/content_cvpr_2018/papers/Mehr_Manifold_Learning_in_CVPR_2018_paper.pdf

(b) Deforming Autoencoders: Unsupervised Disentangling of Shape and Appearance: https://www.ecva.net/papers/eccv_2018/papers_ECCV/papers/Zhixin_Shu_Deforming_Autoencoders_Unsupervised_ECCV_2018_paper.pdf

(c) Rate-Invariant Autoencoding of Time-Series: https://ieeexplore.ieee.org/abstract/document/9053983

In (a), group invariant features are learned in an unsupervised fashion, and a group-invariant loss function is used. Similar ideas have been applied for rotation-invariant autoencoders for spherical images as well. In (b) and (c), the invariant feature and the group transformation are disentangled in an autoencoder which is closely related to this submission. In (b) and (c), the ideas are applicable to much larger groups than considered in the present submission as well -- diffeomorphisms -- which I believe cannot be handled by the method presented submission, at least for now.

Given these works, I think the authors should significantly improve the related work section. They should put this submission in context of such works and better explain the main advantages their method provides.

2. Missing comparison with all these related work: I do think that the papers I have cited and the ideas therein are closely related to the spirit of this work. And thus, the authors should provide some experimental comparisons to one or more of these works.

3. MNIST and Tetris are toy datasets in my opinion. At least some experiments with more complex datasets like CIFAR-100, Patch Camelyon or ShapeNet are needed to better validate the effectiveness of the method. Authors do not provide quantitative measures of reconstruction quality for all experiments, which makes reproducibility hard. Also, non-equivariant baselines and corresponding quantitative evaluation are useful to compare the proposed method and to find out whether indeed the group invariant and equivariant network can be significantly smaller for the same reconstruction quality.

Overall, I think there are too many weaknesses in this submission and am recommending a rejection.


## Post author response

I think the authors have addressed all my questions well. They have also addressed the weaknesses pointed out by the other reviewers well. The additional experiments and comparison with the QAE baseline make the contributions of the paper stronger.

I hope the authors will expand their related work section and put the work in context of several other closely related papers (I only mentioned three, but there are others). I don't fully agree that because some of the papers can also model local transformations and this submission can handle only global transformations, that these papers are not closely related. In fact, they point to some limitations of the current work. For example, how can this submission be extended to handle infinite-dimensional groups which are important in image and time series registration? I am now recommending a weak accept.

---

> ### Author Response · Authors · 2022-08-02
> **Response 1: Missing Related Work**
>
> We thank the reviewer for their valuable comments. Below you will find our responses to your comments.
>
> We thank the reviewer for making us aware of missing related work. We will rework the related work section and also include the works mentioned by the reviewer.
> Especially we will additionally discuss our work in the context of related work that either utilizes approximated group invariant representation learning or defines a group invariant loss function by iterating/optimizing over all group elements at every back propagation step. We take here the opportunity to clarify the relation of the mentioned works (a-c) to our submission:
>
> - a) Manifold Learning in Quotient Spaces: Indeed, this referenced work is closely related to our work. They propose an autoencoder that maps orbits of elements (quotient space) to an invariant representation and decodes them back to the input space - similar to our work. As discussed in both our and their manuscript, such an autoencoder can only efficiently be trained if the encoded and decoded elements are co-aligned. They approach this issue by minimizing an adapted reconstruction loss $\bar{d}(x, y) = \text{inf}_{h\in G}d(h.x,y)$. Specifically, during training they have to iterate over all group elements $h\in G$ to find the one that minimizes the reconstruction loss (bringing x and y in the same pose). While this is feasible for (small) finite groups, for continuous groups they either have to approximately discretize them or perform a separate optimization of $h$ at every back propagation step to find the best match. For example, they descritize the continuous group SO$(2)$ by only considering a subset of 36 rotations. This is of course only an approximate solution to the alignment problem and hence the resulting loss is only approximately group-invariant.
> In contrast, we propose to learn to predict the group element directly from the input, effectively parameterizing $h$ and optimizing it along the encoder and decoder model, minimizing the reconstruction objective (compare line 176 in our updated manuscript). By utilizing proper group-equivariant functions, the resulting loss is perfectly group-invariant and not only approximately. We included this discussion in the related work section.
> Moreover we compared our model against the approach proposed in (a) and demonstrate how our proposed model results in more expressive representations (see below).
>
> - b) Deforming Autoencoders: This work is indeed
> related to ours regarding the idea to disentangle the latent space in an unsupervised learning framework. The fundamental difference consists in the fact that we consider *global* transformations
> on the input space (rotations, translations, etc), and the corresponding embedding is split into
> a shape part and a global transformation action part. Reference (b) instead considers *local*
> deformations, so that the embedding is split into an appearance and a deformation part. Another main difference with our method is that, moreover, their approach is not manifestly equivariant with respect to such deformations. Their networks do not learn representations of the group of local deformations, but rather learns an approximate deformation in such a way the reconstruction task is successful.
> That is, such approach needs to see various deformation of the same sample
> in order to learn to correctly reconstruct (hence, it still needs augmentation), while our explicit equivarint approach needs just one sample per orbit in order to learn the reconstruction.
> One potential connection between theirs and our work would be applying our formalism to local or gauge groups. Loosely speaking, a gauge group $G$ is a group that acts locally, that is, its elements $g\in G$ depends on the space coordinates, that is, $g = g(x)$. These groups are very relevant in mathematics and especially in physics, where the Lagrangian of a system always admits invariance with respect to gauge groups (potentially trivial). One example is the Schrödinger equation, which is invariant with respect to a U$(1)$ gauge symmetry. Physically, this mean that in quantum mechanics the theory is invariant to the phase of the wave function. Even if it would be out of scope for the current work, it would be compelling to explore this further, and extend our formalism to local/gauge groups. Such models could be potentially useful to learn, for example, quantum mechanical or physically-relevant embeddings.
>
> - c) Rate-Invariant Autoencoding of Time-Series: This work is similar in spirit to the reference (b),
> as the authors' goal, in the context of time series, is to obtain an embedding which is rate-invariant. Their AE strategy is then to learn a splitting of the embedding in a rate-invariant and a rate-variant part, in such a way that the reconstruction task can be achieved. Again, this approach focuses on local transformation of the data, as opposite to ours, which focuses on global transformations.

---

> ### Author Response · Authors · 2022-08-02
> **Response 2: New experimental evaluations**
>
> As mentioned above, we implemented and trained a Quotient autoencoder (QAE) proposed by Mehr et al. (ref (a) mentioned by reviewer) on the MNIST dataset for the group $\text{SO}(2)$, discretized in 36 rotations with the loss
>
> $
>     \text{min}_{\theta \in \{10i, i=0,\dots, 35\}} \text{MSE}(x - \rho_X(g(\theta))y)~,
> $
>
> where $x$ is a MNIST sample and $y$ is the reconstructed sample.
>
> We evaluated the resulting embeddings on the rotated MNIST test set (in such a way that the evaluation is the same as for our model). Note, that this test set comprises more than just 36 rotations. We computed TSNE embeddings for this approach (compare Appendix B in revised manuscript), and we can observe that the embedding space shows a clearer structure, in comparison with the classical model. However, in comparison, our approach results in a better clustering of the different digits classes. That shows that the discretization step, while it helps in structuring the embedding space in ``signal clusters'', still does not capture the full continuous nature of the group.
>
> To further quantitatively compare the three methods (ours, QAE (a) and classical AE), we evaluated the reconstruction loss as well as the (digit class) classification accuracy of a KNN classifier trained on 1000 embeddings of each method. We present in the table below the results for the reconstruction loss and for the classification accuracy of a KNN classifier trained on the AE embeddings. To obtain a fair comparison, we kept the architecture and the training hyperparameters exactly identical for all the strategies. We note that our strategy outperforms both the classical AE as well as the strategy of (a) in both tasks.
>
> | Model      | Rec. Loss | KNN Acc.
> | ----------- | ----------- | ----------- |
> | classical      | 0.0170       | 0.68
> | QAE   | 0.0227        | 0.82
> | ours   | 0.0162       | 0.90
>
> We will include this comparison and discussion in the updated manuscript. Moreover,  as suggested by the reviewer, we will include reconstruction accuracies of the other experiments to the manuscript as well. We thank the reviewer for suggesting to discuss and compare our work to this related work, strengthening our manuscript.

---

> ### Author Response · Authors · 2022-08-02
> **Response 3: Questions**
>
> The reviewer writes:
> >The authors should discuss what, if any, are the downstream applications of such a method. Is it data compression? Or do they envision the group invariant features learned by their method to be useful for high-level tasks like robust classification?
>
> Indeed, the learned group invariant latent space can be used for robust (supervised) learning tasks of invariant properties (e.g. ground state energy of a molecular conformation), where in a low-data regime, training a classifier from scratch bares the risk of poor generalizability due to data scarcity of training labels (e.g., see our new experiment on MNIST downstream classification).
>
> Another interesting use case for our proposed unsupervised framework is the pretraining of powerful feature extractors that can later be fine-tuned on classification/regression tasks. We performed additional experiments on the pretrained group-invariant AE on the extended GEOM-QM9 dataset which, as opposed to the standard QM9 dataset ($\sim 130k$ samples), contains multiple conformations of small molecules. We trained the autoencoder on a reduced set of GEOM-QM9 ($\sim 641k$), containing up to $10$ conformations per molecule and utilized this pretrained encoder network to regress (invariant) energy targets, such as internal energy $U$ or enthalpy $H$ on the original QM9 dataset.
>
> We observed that the pretrained encoder network learns faster and achieves better generalization performance than the architectural identical network trained from scratch. In Appendix D of the revised manuscript we  illustrates the learning curves for the two networks on different fraction on $5\%$ and $25\%$ labelled samples from original QM9 dataset to analyze the benefit of finetuning a pretrained encoder network on a low-data regime, when regressing on $H$. On a held-out test dataset of $1000$ samples, the pretrained encoder network achieves superior generalization performance in terms of $R^2$ with $0.7529$ vs. $0.0970$ in the $5\%$ data regime, and $0.9908$ vs. $0.9093$ in the $25\%$ data regime compared to the encoder that was trained from scratch.
>
> Another useful application of our framework is generative modelling in the group invariant latent space. The proposed method can easily be transformed to a Variational Autoencoder which can be used to generate samples conditioned on an invariant representation.
>
> We will include a more in-depth discussion of potential downstream applications of our proposed method to the manuscript. We thank the reviewer for suggesting this.
>
> The reviewer also writes:
> >Related to reference (a) above, if the eventual goal is to use these group-invariant features for tasks like classification, I wonder why we need to find the group action at all. If we have a group-invariant latent feature, we can use a decoder and a group invariant reconstruction loss for training the autoencoder.
>
> As discussed above, in general it is not trivial to define a group-invariant loss. While this might be possible, following (a), for discrete groups, for continuous groups one can define only an approximately invariant loss. This will lead to worse group-invariant representations (compare our new experiment where we compare our method against the method proposed in Mehr et al. (a)).
>
> Another option for defining an invariant loss is to align/match the encoded and decoded elements by an additional algorithm before calculating and back-propagating the reconstruction loss. We argue that this can get computationally infeasible quickly for large discrete groups (e.g. optimal assignment to match two graphs) or continuous groups. To tackle this issue, we propose to learn to predict the group action to solve the co-alignment problem in the reconstruction loss as a non-approximated and efficient way.

---

> ### Author Response · Authors · 2022-08-04
> **Response 4: New Experiment on ShapeNet**
>
> The reviewer shares their concern about the use of toy data sets like MNIST and Tetris. We believe that our contribution is mainly in proposing a novel framework for unsupervised invariant representation learning which we think is best demonstrated on easy to interpret datasets. Moreover we demonstrate the performance of our proposed framework on the complex, real world dataset QM9 of molecular conformation. Still, we understand this concern, and followed the reviewers advise, running **additional experiments on the ShapeNet dataset**. As the dataset comes in an aligned form (e.g. cars are always aligned in the same orientation), we additionally applied random 90 degree rotations to remove this bias.
>
> We utilized 3D Steerable CNNs proposed by Weiler et al. (2018) as equivariant encoder for the 3d voxel input space. We utilized the scalar outputs as rotation-invariant embedding ($z$) and predict (analogously to our experiments on 3d point clouds) 2 rotation-equivariant vectors to construct a rotation matrix ($g$). Similar to our MNIST experiment, we compared the resulting embedding space to the embeddings produced by a non-invariant autoencoder model. We again uploaded a revised manuscript where we visualize in Appendix F a TSNE projection of the embeddings of both models. We can see a well structured embedding space for our model with distinct clusters for the different shape classes. On the other hand, the embeddings produced by the non-invariant autoencoder is less structured and one can make out different clusters for the same shape label but in different orientations.
>
> Moreover, we compared the downstream performance and generalizability of a KNN classifier on shape classification, trained on 1000 embeddings and tested on the rest. The classifier based on our rotation-invariant embeddings achieved an **accuracy of 0.81** while the classifier based on the non-invariant embeddings achieved an accuracy of only 0.63.
>
> We hope that this additional experiment on another complex dataset demonstrates the effectiveness and broad applicability of our method, as suggested by the reviewer. We will add this experiment to main part of the paper in final manuscript. We thank the reviewer for suggesting this additional experiment, as we think it is a strong demonstration of the performance of our proposed method.

---

> ### Author Response · Authors · 2022-08-08
> **Response to Revised Review**
>
> First of all, we thank again the reviewer for suggesting relevant related work and for suggesting the additional experiments. Here we briefly address the additional comments from the revised review.
>
> Regarding the additional references, we do think this is related work to ours and it will be included in the revised version of the paper. In our previous answer we mostly focused on the elements where these work differ from ours to highlight the novelty of theirs and our approach. Nonetheless, it is definitely related to our work, and we thank again the reviewer for point these out to us.
>
> We wish, however, to emphasise once more one very crucial aspect of our work that differs from the above cited work (b). In their approach, the authors define two functions that during training learn separate aspects of the images. While these two functions are approximately related to shape and deformation of an image, they do not satisfy the strict (group-theoretic) definition of invariant and equivariant, respectively, embeddings for the local group of deformations. In other words, their AE is only approximately equivariant with respect to deformations, that is, a deformation of the input image will only be approximately equivariant. In our framework, the AE is *exactly* equivariant with respect to the group transformation of the input.
>
> Finally, our framework can be extended to local (infinite) groups as well. An example would be the case of the group of dilations, that is, an element $g\in G$ act as a (point dependent) scaling factor. That is, if $f$ is a $V$-valued function, where $V$ is a vector space (for example $\mathbb{R}^2$ for 2D images or $\mathbb{R}^3$ for 3D point clouds), we have
>
> $ \rho_X(g)f(v) = \lambda(v)f(v), \text{ for all } v\in V~.$
>
> In fact, this group has infinite dimension since it is defined by the parameters $\lambda(v)$, where $v\in V$, which is
> often infinite dimensional. (More precisely, $\lambda(v)\in G$ is a section of a principal $G$-bundle over $V$.)
> To apply our framework, we would need to apply the same procedure of parameterizing a group action $\psi$, but in
> this case this will also depend on $v$, that is, $\psi = \psi(v)$.
> Practically, we would need to discretize the space (like a grid for image data or temporal intervals for time series), and predict a group action for all the points in the discretized space.

---

> ### Author Response · Authors · 2022-08-09
> **Response to Revised Review 2: Additional Related Work Section**
>
> We wish again to thank the reviewer for encouraging us to expand the related work section.
>
> We updated our manuscript with a preliminary additional related work section (currently in Appendix G due to the current page limit). Please note that this is still a work-in-progress and we are actually working on it and include further relevant literature, but since the deadline for updating the draft is today, we decided to upload it in its current form, to show our commitment to actually follow the reviewer's advice.

---

### Official Review · Reviewer_E8UY · 2022-07-11

**Rating:** 7
**Confidence:** 3
**Soundness:** 4 excellent
**Presentation:** 3 good
**Contribution:** 4 excellent

**Summary:**

Designs an autoencoder that projects each input element into a pair of vectors: one that is group-invariant, and one that represents the group action necessary to recover the original vector from the decoded group-invariant representation.  The group action encoder is designed as a learnable map followed by a deterministic map; the consequent of proposition 2.7 is a sufficient condition to guarantee that this construction results in a suitable group function.  Learnable maps are proposed (and proven suitable) for 2d and 3d rotation groups, permutation groups, and translation groups.

**Questions:**

The notation is challenging, and its challenges are exacerbated by typos and undefined terms.  For example:

GL(X) is never defined.

On line 82 of p. 3, the application of f_X to g and x is written as f_X(g)x, but in equation (1), it is written as f(\rho_V(g^{-1})x) -- I think that in both cases, the x should be inside the parentheses.

A map \phi is said to be G-equivariant if \psi(...) = ...\psi -- either the \phi should be a \psi or vice-versa

we wish to learn the invariant map \phi_{inv}, thus... --- but \phi_{inv} is never mentioned in the definition on the next line

Just in general, I didn't understand the purpose of the \rho_X(g) notation.  If I had to guess, I would say that it takes a general representation, g, of a member of the set G, and converts it into the instantiation of that member that is applicable to external dataset X?  The definition of this notation in the text is self-contradictory and unhelpful: a group action is defined as f:G,X -> X, then a representation is defined as (1) a group action, but also (2) as a function f:X -> GL(X), which seems to be a contradiction.  Neither (1) nor (2) seems to fit with the way \rho_X(g) is used in the rest of the article.



**Limitations:**

Limitations and ethical implications are not discussed.

**Strengths And Weaknesses:**

Experimental examples show that the proposed encoder results in group-invariant representations with very high classification accuracy and very high cluster purity for rotation, permutation, and rotation+permutation+translation groups.

Obviously, many other papers have proposed group-equivariant and group-invariant autoencoders.  This paper seems, to me, to go significantly beyond those previous proposals, because the consequent of Proposition 2.7 provides a sufficient condition for the design of an autoencoder with separable group-invariant and group-action representations.

It's interesting that Prop. 2.7 says that if mu is G-equivariant, then G_x=G_\mu, but only the consequent is necessary in Prop. 2.9 to prove that \xi\circ\mu is a suitable group function.  As a result of affirming the consequent in this way, we never get a proof that a suitable group function is also G-equivariant.  It seems intuitively to be the case, but it is not proven, nor, perhaps, necessary.

---

> ### Author Response · Authors · 2022-08-02
> **Response**
>
> ### General Response
>
> We thank the reviewer for their valuable comments and the encouraging assessment of our work. Below you will find our responses to your comments.
>
> ### Notation
>
> We thank the reviewer for spotting some typos/missing definition in the text. We have corrected those in the revised version.
>
> We just point our here that the maps $f_X$ and $f$ are different maps. The first represent a group action $f_X: X, G \rightarrow X$ or, equivalently,
> $f_X: G \rightarrow \text{GL}(X;n)$, where $\text{GL}(X;n)$ is the general linear group, defined as the group of $n\times n$ invertible matrices with the matrix multiplication operation, and $n$ is the dimension of the vector space $X$.
> The second, which we use in eq. (1), is a map between vector spaces $f:V\rightarrow W$. Thus, for the notation, for $g\in G$, we write $f_X(g)x$, since now $f_X(g)$ is a matrix acting on a vector space, thus we do not need additional parenthesis.
>
> We noticed however, that the notation is prone to confusion given the similarity of the symbols. Thus, we replaced in the revised text $f_X$ with $\rho_X$, in accordance with the notation used in the remainder of the text.
>
> ### Group action vs. Representations
>
> Here we try to motivate the formalism behind the notation used to represent representations acting on vector space.
>
> First, we noticed a serious typo in the main text, which may have caused difficulties in understanding our notation. In the text, we erroneously wrote
> $\rho_X: X \rightarrow \text{GL}(X)$ instead of the correct $\rho_X: G \rightarrow \text{GL}(X)$. That is, a representation is a map that maps a group element $g\in G$
> to an invertible $n\times n$ matrix (assuming $X$ is a $n$-dimensional vector space). This notation is consistent with our usage of it in the main text, as we represent data (or features) as points in the corresponding vector spaces, and we use the corresponding representations in order to apply group transformations.
>
> Equivalently, a representation of the form $\rho_X: G \rightarrow \text{GL}(X)$ can be seen as a group action of the form $\rho : G, X \rightarrow X$, as follows:
>
> $
> \rho(g,x) = \rho_X(g)x~.
> $
>
> Thus, for vector spaces, the two formalism are equivalent.
>
> In the main text, we mentioned with the group action formalism before restricting to representation on vector spaces, as they can be formally also applied to more general spaces (not necessary with a linear structure like vector spaces). Additionally, we made clear in the main text that we restrict to vector spaces, so the representation formalism is justified.

---

### Official Review · Reviewer_CwTC · 2022-07-21

**Rating:** 6
**Confidence:** 4
**Ethics Flag:** Yes
**Soundness:** 2 fair
**Presentation:** 3 good
**Contribution:** 2 fair

**Summary:**

This paper proposes an invariant and equivariant unsupervised learning method, i.e., an invariant and equivariant autoencoder.
When considering such a problem, the naive problem is the decoder construction.
This is because when the invariant model maps to an intermediate space, the group action is already obvious in that space, and the information about the group has disappeared.
The method used in this paper avoids this problem by adding group labels to the intermediate space, and from their experimental results with rotated MNIST and others, it appears that this method indeed achieves a better representation than the usual auto-encoder.

**Questions:**

See Weaknesses.

**Limitations:**

Not applicable.

**Strengths And Weaknesses:**

Strengths
This paper is based on a natural conception and solves the problem of invariant auto encoders.

Weaknesses
The proposed method and solution are too natural to be non-trivial.
It would be better if situations in which expressions created in this way would be useful could be shown.

---

> ### Author Response · Authors · 2022-08-02
> **Response**
>
> We thank the reviewer for their valuable comments. Below you will find our responses to your comments.
>
> Regarding the comment that the reviewer makes, stating that our solution is "too natural to be non-trivial", we wish to re-iterate how our methods works, and why it is actually highly non-trivial.
>
> We wish to learn a (decoder) map from an invariant bottleck to a non-invariant representation of the data. Now, since the bottleneck is invariant, different poses of the same shape will be decoded to the same (from encoder unknown) shape. Let us call this the decoder-canonical-shape.  In order to enable the model to complete the reconstruction task, we learn an additional function $\psi$ (the group function), whose task is to predict the group element corresponding to the transformation between the input shape and the decoder-canonical-shape.
> This is indeed non-trivial since the point of $\psi$ is not learning an absolute pose (which would be indeed trivial because of equivariance) but a relative pose with respect to the reconstructed sample. The equivariance property merely ensures that this needs to be learned once per orbit (and not separately for each elements of the orbit).
>
> Specifically:
> - The decoder is unaware of the input pose, since it only has access to the invariant bottleneck embedding.
> - The (full) encoder (including the group function $\psi$) is unaware of the decoder-canonical pose, since this is obtained later in the network.
>  - The only connection happens during training through the loss $d(\rho_X(\psi(x))\delta(\eta(x)), x)~,$ where $\psi$ must learn how the decoder canonically reconstructs orbits.
>
> Thus, it is only through the learning process that $\psi$ can learn how to properly transform the input to match the decoder canonical choice.
>
> The other non-trivial contribution of our work is concerns how to design such a group function $\psi$ given *any* group $G$, both discrete and continuous.
>
> - First, we derive sufficient conditions that such a map needs to satisfy in order for the learning process to be possible. Again, we do not make any assumptions on the group $G$ or on the data space $X$. This is summarized in Propositions 2.3, 2.4 and Lemma 2.5, whose proofs can be found in the appendix. These statements are highly non-trivial, for instance, the fact that $\psi$ needs to be equivariant up to the stabilizers of all the points of $X$.
>
> - Second, we propose an explicit construction which makes use of the two maps $\mu$ and $\xi$ and of the intermediate space $Y$. This construction satisfy all the requirement and it is group-agnostic. This is described in Proposition 2.6. Again, the design through an intermediate homogeneous space is very non-trivial, and its applicability is completely general, as we show in Propositions 2.8, 2.7, 2.9, as well as in the examples in Section 3.
>
> #### Additional Experiments
>
> Another non-trivial aspect of our work is that it is applicable on real-world dataset, as the QM9 dataset in section 5.3, where the model is able to reconstruct to a very high degree of accuracy conformations (3d coordinates) of small molecules.
>
> In this context, the learned group invariant latent space can be used for robust (supervised) learning tasks of invariant properties (e.g. ground state energy of a molecular conformation), where in a low-data regime, training a classifier from scratch bares the risk of poor generalizability due to data scarcity of training labels (e.g., see our new experiment on MNIST downstream classification in Appendix B of the revised manuscript).
>
> We performed additional experiments on the pretrained group-invariant AE on the extended GEOM-QM9 dataset which, as opposed to the standard QM9 dataset ($\sim 130k$ samples), contains multiple conformations of small molecules. We trained the autoencoder on a reduced set of GEOM-QM9 ($\sim641k$), containing up to $10$ conformations per molecule and utilized this pretrained encoder network to regress (invariant) energy targets, such as internal energy $U$ or enthalpy $H$ on the original QM9 dataset.
>
> We observed that the pretrained encoder network learns faster and achieves better generalization performance than the architectural identical network trained from scratch. In Appendix D of the revised manuscript we  illustrates the learning curves for the two networks on different fraction on $5\%$ and $25\%$ labelled samples from original QM9 dataset to analyze the benefit of finetuning a pretrained encoder network on a low-data regime, when regressing on $H$. On a held-out test dataset of $1000$ samples, the pretrained encoder network achieves superior generalization performance in terms of $R^2$ with $0.7529$ vs. $0.0970$ in the $5\%$ data regime, and $0.9908$ vs. $0.9093$ in the $25\%$ data regime compared to the encoder that was trained from scratch.

---

### Author Response · Authors · 2022-08-04
**General Response**

We wish to thank all reviewers for their insightful comments. We addressed the specific concerns in detailed responses to each of the reviews below. Here, we wish to summarize the main points that might be relevant for all the reviewers and the overall discussion, highlighting the main differences from the original draft and the revised one.

Motivated by the reviewers to apply our method to further complex dataset and to perform further analysis, we conducted the additional experiments, which can be found in the Appendix of the revised draft:

- **ShapeNet with group $G=\text{SO}(3)$**. We showed that our AE can successfully reconstruct rotated ShapeNet samples. We show that it clusters different shapes in the embedding space much better than a classical AE. We show the superiority of our embeddings by training a KNN classifier on them, achieving a $\sim30$% performance gain in comparison to the classical version.

- **QM9 regression with** $G=${$S_n, \text{SO}(3)$} **as pretrained embeddings**. We finetuned the (newly pretrained on GEOM-QM9) AE on a reduced set of QM9 geometries, showing that the pretrained network outperforms the one trained from scratch on property prediction tasks.

- **Comparison with Mehr et al. (2018)**. We compared our approach with the one of Mehr et al. (2018)
    on the rotated MNIST dataset. Our approach is superior both in term of reconstruction loss as well as in the accuracy of a KNN
    classifier trained on the bottleneck embeddings.


In summary, we think that by

- clarifying the main motivation and contribution of our work,

- addressing unclarities in the notation,

- adding a discussion of missing related work and their relation to our work,

- adding additional experiments to compare our work to such related work,

- adding quantitative measures of the (superior) performance of our method (compared to other work) and

- adding additional experiments on more complex datasets,

we address the main concerns of the reviewers and hope that the reviewers reconsider their evaluation of our work.

---

### Author Response · Authors · 2022-08-09
**General Response after Rebuttal**

We take again the opportunity to thank all the reviewers for the insightful reviews, comments and discussion.

We have uploaded a new updated version of the paper with an expended related work section. This can be currently found in Appendix G to respect the current page limit, but will be moved in the main section of the final version. We point out that this section is still in a preliminary form, as we are working on including further relevant literature, but since the deadline for
updating the draft is today, we decided to share it anyway in its current form.

Additionally, we performed an additional experiment (MNIST with $G=\text{SO}(2)$), where we compared our model to a fully equivariant AE (that is, its embedding is equivariant), where we extracted **post-training** an invariant embedding through an invariant pooling of the equivariant one. While the reconstruction loss of the fully equivariant embedding is comparable to ours, the corresponding invariant embeddings are, as claimed, less informative, as we show in the updated draft (Appendix B) through TSNE plots as well as the accuracy of a KNN classifier, which performs at best on par with a classical model.

We believe that the additional experiments and discussions substantially improved our paper (and we thank again the reviewers for the insightful suggestions!), and we would appreciate if the reviewers were willing to reconsider their evaluation to reflect such improvements.

---

### Meta-Review · Area_Chair_QeTw · 2022-08-25

**Recommendation:** Accept
**Confidence:** Certain

**Metareview:**

The paper proposes an auto-encoder that maps a point to a canonical point and  to a group element such that the composition of the group and the canonical point reconstructs the point (the invariance / equivariance is in this sense).   The method is promising since it learn in an unsupervised way the group actions. Experiments were on simple tasks and group actions. Reviewers were overall positive and the paper improved a lot during rebuttal/ revision thanks to their recommendations.

I have one recommendation to the authors to include the use of the representation learned in a classification task to see how the learned representation alleviates the need of large training samples.

Accept


**Award:**

No

---

### Decision · Program_Chairs · 2022-09-14

Accept